# A Comparative Analysis of China's Anthropogenic $CO_2$ Emissions (2000–2023): Insights from Six Bottom-Up Inventories and Uncertainty Assessment

Huirong Yang[1,2,3], Kai Wu[1,3], Huizhong Shen[4,5], Greet Janssens-Maenhout[6], Monica Crippa[6], Diego Guizzardi[6], Minqiang Zhou[1,3]

[1]Key Laboratory of Atmospheric Environment and Extreme Meteorology, Institute of Atmospheric Physics, Chinese Academy of Sciences, Beijing, 100029, China
[2]University of Chinese Academy of Sciences, Beijing, 101408, China
[3]LAGEO & CNRC, Institute of Atmospheric Physics, Chinese Academy of Sciences, Beijing, 100029, China
[4]Guangdong Provincial Observation and Research Station for Coastal Atmosphere and Climate of the Greater Bay Area, School of Environmental Science and Engineering, Southern University of Science and Technology, Shenzhen, 518055, China
[5]Shenzhen Key Laboratory of Precision Measurement and Early Warning Technology for Urban Environmental Health Risks, School of Environmental Science and Engineering, Southern University of Science and Technology, Shenzhen, 518055, China
[6] European Commission, Joint Research Centre (JRC), Ispra (VA), 21027, Italy

*Correspondence to*: Kai Wu (kwu@mail.iap.ac.cn) and Minqiang Zhou (minqiang.zhou@mail.iap.ac.cn)

**Abstract.** Accurate quantification of anthropogenic $CO_2$ emissions is crucial for mitigating climate change and verifying emission reduction policies. This study conducts a comparative analysis of China's anthropogenic $CO_2$ emissions for the period between 2000 and 2023 based on six widely used bottom-up inventories at their latest version (ODIAC2023, EDGAR2024, MEIC-global-$CO_2$ v1.0, CAMS-GLOB-ANT v6.2, GEMS v1.0, and CEADs). The national total $CO_2$ emissions increase from 3.43 (3.21–3.63) Gt year$^{-1}$ in 2000 to 12.03 (11.35–12.98) Gt year$^{-1}$ in 2023, with three growth periods: rapid growth (2000–2013, 0.56±0.013 Gt year$^{-1}$), near-stagnation (2013–2016, -0.07±0.022 Gt year$^{-1}$), and renewed growth (2016–2023, 0.30±0.016 Gt year$^{-1}$). Emissions are dominated by the electricity and heat production, and the industry and construction (78% of total emissions), with the former replacing the latter as the largest source after 2017. EDGAR consistently reports the highest national $CO_2$ emissions, while MEIC provides the lowest, contributing to the large deviations after 2012. EDGAR and MEIC report different spatial distributions of the transport sector. EDGAR concentrates emissions along major roads and MEIC distributes them more diffusely. Extreme outliers (>$10^5$ ton $CO_2$ km$^{-2}$ year$^{-1}$, against an average of $10^2$ ton $CO_2$ km$^{-2}$ year$^{-1}$) in these inventories arise from discrepancies in point source data in the Carbon Monitoring for Action (CARMA) versus the China Power Emissions Database (CPED). Overall, the uncertainty of total national anthropogenic $CO_2$ emissions is within 5% (1σ), and the uncertainties are about 10–50% (1σ) at the provincial level.

# 1 Introduction

The global mean temperature in 2024 was 1.5°C above pre-industrial levels, making it the warmest year in the 175-year record of observations (WMO, 2025). This increases the urgency of achieving the Paris Agreement's goal of limiting global warming to a maximum of 1.5°C (Schleussner et al., 2016). Atmospheric carbon dioxide ($CO_2$) is the dominant greenhouse gas (IPCC, 2017), and its concentration (430.5 ppm in May 2025) is now 1.5 times higher than pre-industrial levels (280 ppm), mainly due to anthropogenic activities (WMO 2024; Etheridge et al., 1996). China, which is responsible for about 80% of East Asia's anthropogenic $CO_2$ emissions (Xia et al., 2025) and about 32% of global $CO_2$ emissions according to the Global Carbon Project (GCP, 2024; available at: https://globalcarbonbudget.org/), has committed to reaching peak emissions by 2030 and carbon neutrality by 2060. Besides, China's energy structure is also undergoing an obvious transition driven by policies such as the renewable portfolio standards (RPS) and the clean air policy, which promote cleaner energy and industrial upgrades. The share of renewables in total power generation increased from 16.6% in 2000 to 28.2% in 2020, although fossil fuels still dominate and overcapacity issues remain (Zhao et al., 2022). Under this ongoing energy transition, accurate quantification of anthropogenic $CO_2$ emissions and understanding the uncertainties in emissions inventories are needed to guide emission reduction policies toward the dual-carbon goals (Li et al., 2017a).

A variety of bottom-up emission inventories have been developed to quantify anthropogenic $CO_2$ emissions based on activity data and emission factors (EFs). The gridded inventories apply spatial proxies to allocate emissions across grid cells (Han et al., 2020a), including point sources (e.g., power plants), line sources (e.g., road networks), and area sources (e.g., population density, gross domestic product (GDP), nighttime lights). Global gridded products provide consistent, worldwide estimates with high spatial resolution (1 km or 0.1°), such as the Open-Data Inventory for Anthropogenic Carbon Dioxide (ODIAC) (Oda et al., 2018; Oda and Maksyutov, 2011), the Emissions Database for Global Atmospheric Research (EDGAR) (Janssens-Maenhout et al., 2019), the Global Emission Modeling System (GEMS) (Wang et al., 2013), and the Copernicus Atmosphere Monitoring Service (CAMS-GLOB-ANT, hereafter referred to as CAMS, Soulie et al., 2024). China-specific inventories use provincial energy statistics and locally optimized EFs to account for national and subnational $CO_2$ emissions, such as the Multi-resolution Emission Inventory for China (MEIC) (R. Xu et al., 2024; Li et al., 2017a; B. Zheng et al., 2018), the China High Resolution Emission Database (CHRED) (Cai et al., 2018) and the China Emission Accounts and Datasets (CEADs) (J. Xu et al., 2024; Y. Guan et al., 2021; Shan et al., 2018, 2020).

Despite the different allocation methods and underlying data, the uncertainties in the overall magnitudes and trends of $CO_2$ emissions between the global inventories are within 10% at the global scale (Oda et al., 2019; Han et al., 2020a; R. Xu et al., 2024). However, at the national scale, the uncertainties can reach 40-100% (Peylin et al., 2013) and can be even larger at the regional and city scales, e.g., 300% in the Beijing-Tianjin-Hebei area (Han et al., 2020b). The uncertainties between the different inventories are caused by three factors. First, different official statistics can lead to large emission gaps (D. Guan et al., 2012; Hong et al., 2017). Previous studies have shown significant discrepancies in energy consumption from different official statistics in China. Provincial-level data tend to align more closely with satellite observations than national-level

statistics (Akimoto et al., 2006; D. Guan et al., 2012; Zhao et al., 2012). Second, the EF is another key element that causes the differences. The IPCC-based EFs used by ODIAC and EDGAR may not correctly reflect the specific fuel quality and combustion technologies in China (e.g., the EF for raw coal in CEADs and ODIAC is 0.499 and 0.746, respectively) (Han et al., 2020a). Third, spatial proxies determine how emissions are distributed across grid cells. For example, relying on outdated point-source databases such as the Carbon Monitoring for Action (CARMA) (the last update was on 28 November, 2012) may incorrectly distribute emissions in urban areas and introduce extrapolation errors (Han et al., 2020a; Wang et al., 2013; M. Liu et al., 2013), while more comprehensive power plant inventories such as the China Power Emissions Database (CPED) provide better spatial accuracy (Li et al., 2017b; F. Liu et al., 2015).

Previous studies have demonstrated large discrepancies among anthropogenic $CO_2$ emission inventories in China and investigated the possible reasons. Han et al (2020a) compared nine global and regional inventories for China and found that differences in activity data and EFs can lead to significant uncertainties in emission estimates, with the maximum difference in 2012 reaching up to 33.8%. Zheng et al (2025) conducted a cross-scale comparison of EDGAR, MEIC, and CEADs and showed that coarse aggregation reduces the impact of outlier emission values, and leads to stronger agreement between inventories at a resolution of $3° \times 3°$ compared to $0.25° \times 0.25°$. At the city level, Liu et al (2024) found that the relative standard deviations between six inventories are more than 50%, with uncertainties showing a strong logarithmic dependence on proxy variables such as population density and nightlight data. In recent years, China has announced a series of policy measures aimed at reducing carbon emissions, alongside changes in factory technology and energy structure. These developments underscore the urgent need for accurate and timely quantification of anthropogenic $CO_2$ emissions. Moreover, emission inventories are continuously updated to incorporate improved inputs (e.g., activity data, EFs, and refined methodology). Therefore, it is crucial to use the latest versions of the various inventories to capture these methodological updates and better understand the most recent patterns of China's anthropogenic $CO_2$ emissions.

To this aim, this study conducts a comprehensive analysis of the spatiotemporal variation of China's anthropogenic $CO_2$ emissions and investigates the differences among six widely used emission inventories at their latest versions: the global inventories ODIAC, EDGAR, MEIC, GEMS, CAMS, and the China-specific inventory CEADs. The data and methods are presented in Section 2. We report our results in Section 3 and conclude the paper in Section 4. Compared with previous studies (Han et al., 2020b; Zheng et al., 2025), we extend the temporal coverage to 2000-2023, enabling a more current and consistent assessment of recent emission trends, inter-inventory discrepancies, and scale-dependent uncertainties across China.

## 2 Data and methods

To ensure both temporal completeness and spatial representativeness, the selected emission inventories must provide a continuous time-series covering most of the 2000-2023 period (with at least 2000–2019 coverage in GEMS) and have explicit spatial coverage over mainland China. Six anthropogenic $CO_2$ emission inventories, including five gridded

inventories (ODIAC2023, EDGAR2024, MEIC-global-$CO_2$ v1.0, CAMS v6.2, and GEMS v1.0) and one urban total emission inventory (CEADs), are applied to provide estimates of total emissions at the national, provincial, and city levels in China. As internationally recognized and widely used by previous studies (Li et al., 2017b; Han et al., 2020b; Liu et al., 2024; Zheng et al., 2025), these inventories are publicly available from official repositories.

In addition to these six datasets, the National Greenhouse Gas Inventory (NGHGI) submitted by the Chinese government to the United Nations Framework Convention on Climate Change (UNFCCC, available at: https://unfccc.int/reports) was also collected. The NGHGI provides the officially reported national total emissions and therefore serves as an independent benchmark for evaluating the reliability of the six inventories. As NGHGI covers only discrete years (2005, 2010, 2012, 2014, 2017, 2018, 2020, and 2021), it is not included in the continuous temporal analysis but is used solely for national-level comparison.

The specific information of the six selected inventories is presented in Section 2.1. Table 1 lists the temporal and spatial resolution, data version, and principal downscaling proxies of those inventories. All five gridded inventories were standardized to a common $0.1° \times 0.1°$ coordinate system and a common unit of ton $CO_2$ km$^{-2}$ year$^{-1}$ (Section 2.2).

Table 1. Specification of emission inventory statistics.

| | ODIAC | EDGAR | MEIC | CAMS | GEMS | CEADs |
|---|---|---|---|---|---|---|
| Version | ODIAC2023 | EDGAR2024 | v1.0 | v6.2 | v1.0 | NA |
| Domain | Global | Global | Global | Global | Global | China |
| Temporal coverage | 2000-2022 | 1970-2023 | 1970-2023 | 2000-2026 | 1700-2019 | 1997-2021 |
| Time resolution | Monthly or annual | Monthly or annual | Monthly or annual | Monthly or annual | Monthly or annual | Annual |
| Activity data | CDIAC, BP | IEA | CESY, IEA, BP | EDGAR, CAMS-GLOB-Ship | NBS, IEA | CESY, NBS |
| Emission factors | IPCC | IPCC | CEADs, national submissions in UNFCCC, IPCC | EDGAR | Literature, on-site measurements | on-site measurements |
| Point source | CARMA | CARMA | CPED | EDGAR | WRI | NA |
| Line source | NA | OpenStreetMa | CDRM | EDGAR | NA | NA |

| | | | | | | |
|---|---|---|---|---|---|---|
| | p and OpenRailway Map | | | | | |
| Area source | Nightlight data | Population density and nightlight data | Population density and land use | Population density | Population density, nightlight data and vegetation density | NA |
| Spatial resolution | 1km×1km, 1°×1° | 0.1°×0.1° | 0.1°×0.1° | 0.1°×0.1° | 0.1°×0.1° | NA |
| Unit of gridded emissions | ton C cell$^{-1}$ month$^{-1}$ | ton $CO_2$ km$^{-2}$ year$^{-1}$ | ton $CO_2$ cell$^{-1}$ year$^{-1}$ | kg $CO_2$ m$^{-2}$ s$^{-1}$ | g $CO_2$ km$^{-2}$ year$^{-1}$ | NA |
| Emission estimates | Global | Global and national | Global, national and provincial | Global and National | Global and national | National, provincial and city |
| Year published | 2024 | 2024 | 2024 | 2023 | 2024 | 2017 |
| Data source | https://db.cger.nies.go.jp/dataset/ODIAC/DL_odiac2023.html | https://edgar.jrc.ec.europa.eu/dataset_ghg2024#p1 | http://meicmodel.org.cn/?page_id=2341 | https://eccad.sedoo.fr/#/metadata/479 | https://gems.pku.edu.cn/data/database | https://www.ceads.net.cn/data/ |
| References | Oda and Maksyutov (2011); Oda et al (2018) | Janssens-Maenhout et al (2019) | R. Xu et al (2024) | Soulie et al (2024); | Wang et al (2013) | J. Xu et al (2024); Y. Guan et al (2021b); Shan et al (2020, 2018) |

*All datasets were last accessed on 19 April 2025.

## 2.1 Emission inventories

ODIAC is a global grid-based $CO_2$ inventory that provides monthly emissions at a high spatial resolution of 1 km × 1 km.
115 Total emissions are derived from the Carbon Dioxide Information Analysis Center (CDIAC), which compiles $CO_2$ estimates

from fossil fuel combustion, cement production, and gas flaring using United Nations energy statistics (Andres et al., 2016; Oda et al., 2018, 2019). These national totals are then spatially allocated for point sources using the CARMA power plant database and for area sources using satellite-based nightlight data. ODIAC does not explicitly map line sources such as road traffic. Although streetlights have been proposed as a proxy for such sources (Oda and Maksyutov, 2011), this approach may over-allocate emissions in brightly lit urban areas relative to rural or low-light regions due to the complexity of actual traffic distribution (Wang et al., 2013). We use ODIAC2023, which covers the years from 2000 to 2022.

EDGAR is developed by the Joint Research Centre (JRC) and the Netherlands Environmental Assessment Agency. It combines national energy balance data from the International Energy Agency (IEA) with sector-specific activity data from sources such as BP plc (formerly the British Petroleum company p.l.c.), the United States Geological Survey (USGS), the World Steel Association, the Global Gas Flaring Reduction Partnership (GGFR), the National Oceanic and Atmospheric Administration (NOAA), and the International Fertilizer Association (IFA). Emissions are calculated using IPCC default EFs and spatially disaggregated using CARMA (point source), OpenStreetMap (line source), and population density and nighttime lights (area sources) (Janssens-Maenhout et al., 2019). We use EDGAR2024, which provides annual and monthly data from 1970 to 2023 at a spatial resolution of $0.1° × 0.1°$.

MEIC is developed by Tsinghua University to estimate global and regional $CO_2$ emissions, with a particular focus on China. Emissions are estimated by integrating activity data from multiple international and local statistics, with 72% of global $CO_2$ emissions estimated based on information from individual countries in 2021. In China, the energy statistics data is obtained from the provincial-level database: China Energy Statistics Yearbook (CESY). Point emissions are allocated using the China coal-fired Power plant Emissions Database (CPED), which includes more than 7600 generating units—approximately 1300 additional small power plants more than CARMA — and has been validated using satellite imagery. MEIC uses the transportation network data from the China Digital Road Network Map (CDRM) to constrain the distribution of vehicle activity as well as population density, GDP, and land use for other sectors (Li et al., 2017a; Xu et al., 2024b). In this study, we use the latest MEIC-Global-$CO_2$ product (v1.0), which provides higher spatial resolution ($0.1° × 0.1°$) and longer temporal coverage (1970-2023) than the MEIC-China-$CO_2$ product (v1.4; $0.25° × 0.25°$, up to 2020). It's noteworthy that although MEIC-Global-$CO_2$ is a global product, its emissions calculations for China continue to rely on local energy statistics (CESY) and emission factors (CEADs), ensuring consistency with domestic data while improving spatiotemporal details.

CAMS is a global inventory developed as part of the Copernicus Atmosphere Monitoring Service project. It builds on EDGAR and integrates several complementary datasets, including the Community Emissions Data System (CEDS) for the extrapolation of the emissions up to the current year, the CAMS-GLOB-TEMPO for monthly variability, and the CAMS-GLOB-SHIP for ship emissions. CAMS provides monthly emissions across 17 emission sectors (e.g., transportation, electricity generation, industry, etc.) at a resolution of $0.1° × 0.1°$ (Soulie et al., 2024). The version used in this study is CAMS-GLOB-ANT v6.2, which covers the period from 2000 to 2026.

GEMS is a global $CO_2$ inventory that is developed as a successor to Peking University $CO_2$ (PKU). It updates the EFs based on the latest literature and on-site measurements, and refines the technology splits in sectors such as road transport. The energy statistics come from the National Bureau of Statistics (NBS) for China and from sub-national datasets for many developed and developing countries. For countries lacking sub-national fuel consumption data, national-level statistics from IEA are used. Emissions are classified into seven sectors (power generation, industry, residential and commercial emissions, transportation, agriculture, and natural emissions) or six fuel/activity types (coal, oil, gas, waste, biomass, and industrial processes). The spatial allocation uses World Resources Institute (WRI) for point sources and combines vegetation density, population density, and nighttime lights for the remaining emissions (Wang et al., 2013). We use GEMS v1.0, which covers the period 1700–2021 with a spatial resolution of 0.1°. However, the version available at the time of our analysis only included data up to 2019, which is therefore the endpoint used throughout our study.

CEADs provides annual $CO_2$ emissions at national, provincial, and city scales. The national and provincial emissions are based on CESY and NBS, respectively. In addition to total $CO_2$ emissions, CEADs provides an energy inventory, a $CO_2$ emission inventory for industrial processes, and EFs. CEADs uses locally optimized EFs derived from extensive sample measurements—such as 602 coal samples and over 4000 coal mines for coal EFs—which are considered more representative of China's actual fuel characteristics than the IPCC-based default values (Shan et al., 2018, 2020; J. Xu et al., 2024; Y. Guan et al., 2021). In this study, we use the national and provincial CEADs datasets from 2000 to 2021.

**2.2 Data preprocessing**

To extract the data, we first used a mask with national boundaries (https://cloudcenter.tianditu.gov.cn/administrativeDivision) to extract the emissions within mainland China for the five global grid-based inventories (ODIAC, EDGAR, MEIC, CAMS, and GEMS). To enable consistent comparison between inventories, all gridded datasets were processed to a uniform spatial resolution of 0.1° × 0.1°, with emission units standardized to ton $CO_2$ km$^{-2}$ year$^{-1}$. Unit conversions accounted for original formats and required area normalization for datasets with grid-cell-based values (e.g., ODIAC: ton C cell$^{-1}$ month$^{-1}$, MEIC: ton $CO_2$ cell$^{-1}$ year$^{-1}$). A stoichiometric factor (44/12) was applied to convert carbon to $CO_2$ where necessary (e.g., ODIAC). Spatial resampling was performed to align with the MEIC coordinate system, using nearest-neighbor interpolation or area-weighted aggregation depending on the original resolution. National totals were taken directly from original reports, except for ODIAC, which was summed from gridded data. At the provincial level, emissions were taken directly from the MEIC and CEADs data, while for the other datasets, estimates for the provinces were calculated using spatial zonal statistics based on standardized administrative boundary masks (https://cloudcenter.tianditu.gov.cn/administrativeDivision).

**3 Results**

**3.1 National total $CO_2$ emissions**

The six bottom-up inventories show a significant increase in total national $CO_2$ emissions from 2000 to 2023 (GEMS to

180 2019, CEADs to 2021, ODIAC to 2022), with average emissions increasing from 3.43 Gt year[-1] in 2000 to 12.03 Gt year[-1] in 2023 (Fig. 1). The differences between the emission inventories become more pronounced after 2012 and diverge in recent years, with the emission range (maximum-minimum difference) and the standard deviation (SD) increasing from 0.41 and 0.14 Gt year[-1] in 2000 to 1.63 and 0.58 Gt year[-1] in 2023. Before 2012, both metrics are relatively stable and low (range < 0.82 Gt year[-1], SD < 0.30 Gt year[-1]). After 2013, however, the range is above 1.03 Gt year[-1] and peaked at 1.64 Gt year[-1] in

185 2021, mainly due to EDGAR reporting the highest emissions versus MEIC reporting the lowest emissions.

To further assess the consistency of the six inventories, we calculate the mean absolute difference (MAD), which is defined as the multi-year mean of annual absolute differences between each inventory and either the NGHGI or the six-inventory mean. Compared with NGHGI, the MADs range from 0.156 Gt year[-1] (CAMS) to 0.835 Gt year[-1] (MEIC). Against the six-inventory mean, the MADs range from 0.12 Gt year[-1] (ODIAC) to 0.449 Gt year[-1] (MEIC). EDGAR reports the highest

190 emissions, which is about 0.370 Gt year[-1] larger than the mean emission. MEIC shows the lowest emission levels, which is about 0.449 Gt year[-1] less than the mean emission. Overall, CAMS exhibits the greatest consistency with the NGHGI, being at least 30% lower than that of the other inventories. In comparison, ODIAC agrees most closely with the six-inventory mean, with an MAD at least 58% lower than the others.

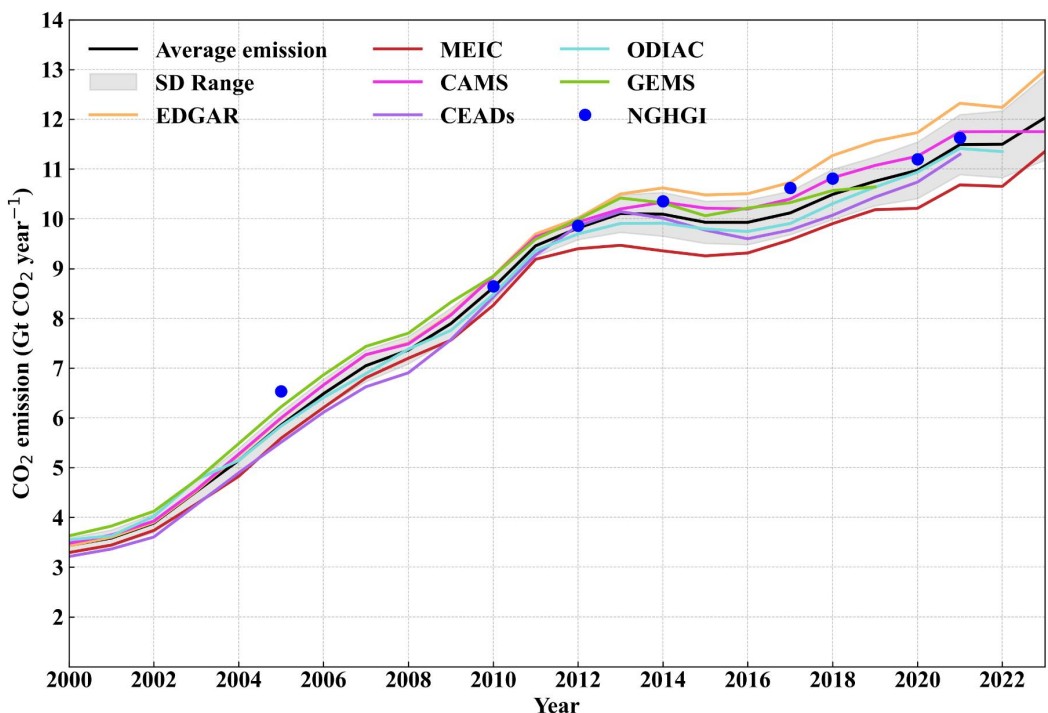

195

**Figure 1.** Annual anthropogenic $CO_2$ emissions in mainland China from 2000 to 2023, as reported by six emission inventories: EDGAR, MEIC, CAMS, CEADs (up to 2021), ODIAC (up to 2022), and GEMS (up to 2019), and one government-reported data (NGHGI). Apart from ODIAC, all inventories provide national totals directly. We calculated China's emissions by summing the grid values within China for ODIAC. The shaded area indicates the standard deviation of the six inventories. It's noteworthy that the inter-inventory mean and SD

200 were calculated from the above mentioned six inventories.

The increase in $CO_2$ emissions shows three different phases (Fig. 1, Table 2). The first phase (2000–2013) shows the most rapid growth, with an average growth rate of $0.56 \pm 0.013$ Gt year$^{-1}$, driven by industrialization, urbanization, and rising energy demand. In contrast, emissions become relatively stable from 2013 to 2016, with all inventories showing a slight decline ($-0.07 \pm 0.022$ Gt year$^{-1}$ on average). This short-term stagnation is mainly influenced by the adjustment of energy structure and industrial upgrades under China's 12th Five-Year Plan, and the implementation of air clean policy since 2013 (Han et al., 2020b; Shi et al., 2022; Zheng et al., 2025). From 2016 to 2023, all inventories show increased $CO_2$ emissions again, with a slower rate ($0.30 \pm 0.016$ Gt year$^{-1}$) compared to the first phase. This rebound could be attributed to the expansion of infrastructure investment and the recovery of coal-based power generation, as the mitigation effect of the cleaner energy mix weakened after 2016 (Zhang et al., 2020).

**Table 2.** Linear regression statistics (correlation coefficient (R) and slope with its uncertainty) between $CO_2$ emissions and year for all six inventories and their average.

| | | Average emissions | EDGAR | MEIC | CAMS | CEADs | ODIAC | GEMS |
|---|---|---|---|---|---|---|---|---|
| 2000-2013 | Slope | 0.56 | 0.58 | 0.53 | 0.56 | 0.57 | 0.53 | 0.56 |
| | Uncertainty of slope | 0.013 | 0.014 | 0.016 | 0.015 | 0.017 | 0.013 | 0.012 |
| | R | 0.99*** | 0.99*** | 0.99*** | 0.99*** | 0.99*** | 0.99*** | 0.99*** |
| 2013-2016 | Slope | -0.07 | -0.01 | -0.06 | -0.01 | -0.19 | -0.06 | -0.09 |
| | Uncertainty of slope | 0.022 | 0.034 | 0.029 | 0.035 | 0.014 | 0.016 | 0.057 |
| | R | -0.91 | -0.26 | -0.81 | -0.23 | -0.99 | -0.93 | -0.74 |
| 2016-2023 | Slope | 0.30 | 0.34 | 0.26 | 0.25 | 0.34 | 0.30 | 0.15 |
| | Uncertainty of slope | 0.016 | 0.024 | 0.023 | 0.024 | 0.027 | 0.024 | 0.022 |
| | R | 0.99*** | 0.98*** | 0.98*** | 0.97*** | 0.99*** | 0.98*** | 0.98* |

Note: *, **, *** denote P<0.05, P<0.01, P<0.001 respectively.

In response to the Paris Agreement's requirement of a global stocktake every five years starting in 2023 (https://unfccc.int/sites/default/files/paris_agreement_english_.pdf), we analyze China's emissions variation every five years, using 2003 as the baseline year corresponding to the first global stocktake (Fig. 2). The highest growth is recorded in the period from 2003 to 2008 (> 0.52 Gt year$^{-1}$) and 2008-2013 (> 0.45 Gt year$^{-1}$), followed by a stable period in the years from 2013 to 2018, in which the CEADs even records a slight decline (-0.01 Gt year$^{-1}$). Growth then resumed in 2018-2023, averaging 0.21 Gt year$^{-1}$.

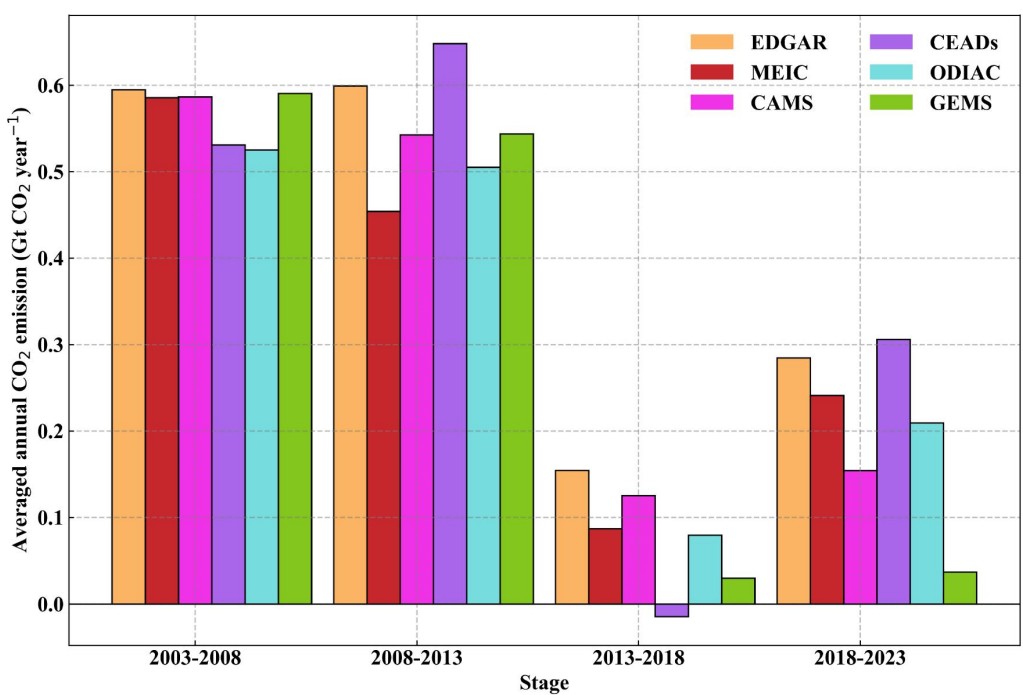

**Figure 2.** Average annual CO₂ emission growth rate during the five-year periods.

We use four major emission sectors defined by MEIC: electricity and heat production, industry and construction, residential and commercial, and transport (Table S1). To ensure comparability, we reclassify the sectoral CO₂ emissions in the other inventories according to this framework (Table S2). The sectoral CO₂ emissions show that the electricity and heat production sector and the industry and construction sector dominate emissions and together account for over 78% of total emissions (Fig. 3). Prior to 2016, emissions from the industry and construction exceeded emissions from the electricity and heat production. However, since 2013, the sector of industry and construction has become stable and even declined in some inventories (MEIC, CEADs, and GEMS), while the sector of electricity and heat production shows a steady upward trend after 2017. As a result, the electricity and heat production became the largest emitting sector in most inventories after 2017 (CEADs: 2016, MEIC and GEMS: 2017, EDGAR: 2018). In addition, residential and commercial emissions as well as the transport sector, show similar trends in most inventories (except GEMS). In most inventories (e.g., EDGAR, MEIC, CAMS, and CEADs), emissions from the residential and commercial sector gradually exceeded those from the transport sector after 2016, while a reverse pattern was observed in GEMS. The residential emissions provided by GEMS are considered more reliable, as the national residential emission survey for the Second National Pollution Source Census was conducted by the GEMS team. Data from their surveys indicate that the publicly available statistical sources (such as the IEA and the Food and Agriculture Organization of the United Nations, FAO) have underestimated the rapid transition of China's residential energy mix (Tao et

al., 2018), leading to likely overestimated residential emissions in other inventories. The changes in the size of sectoral $CO_2$ emissions indicate the changes in China's energy structure and economic growth, highlighting the importance of incorporating locally based surveys for residential emissions to improve the accuracy of bottom-up inventories

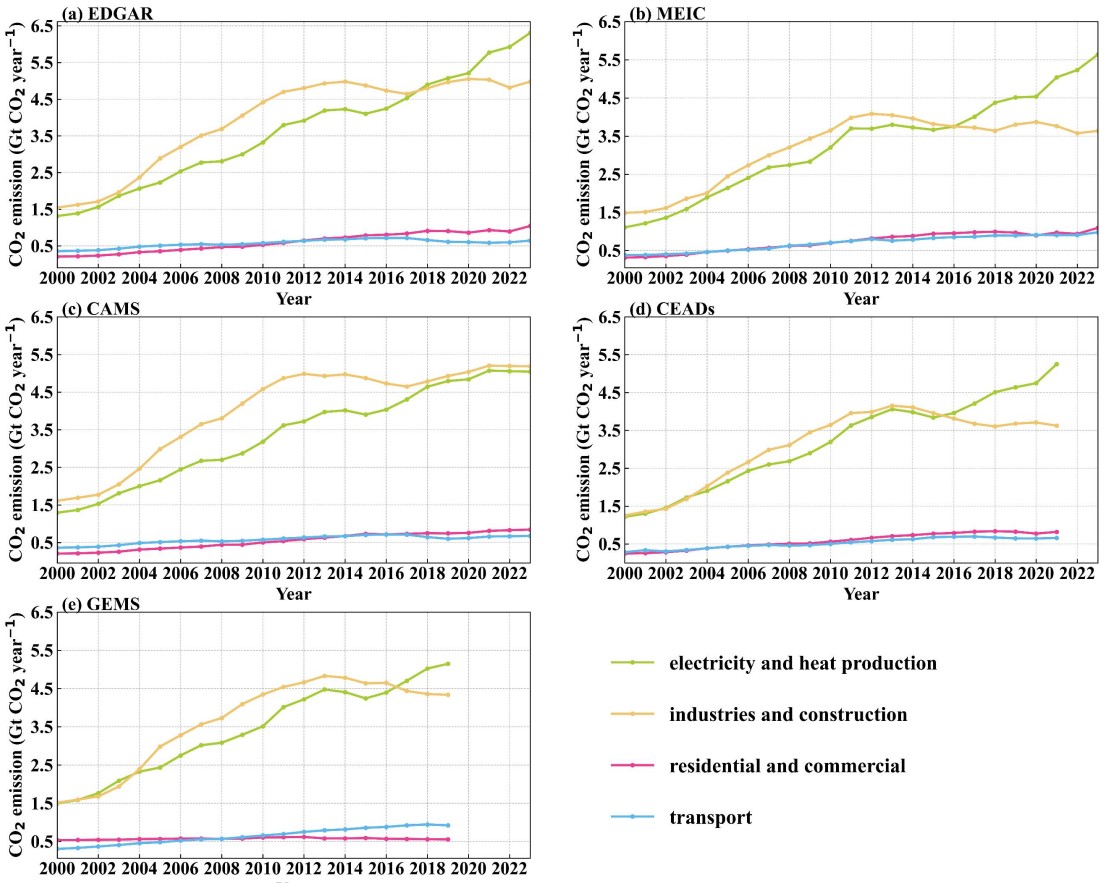

**Figure 3.** Anthropogenic $CO_2$ emissions by sector — electricity and heat production, industries and construction, residential and commercial, and transport—for the period 2000–2023, as reported by EDGAR (a), MEIC (b), CAMS (c), CEADs (d), and GEMS (e). Although CEADs provides both national- and provincial-level sectoral data, the national-level version is used here for consistency with other inventories. ODIAC does not provide sectoral $CO_2$ emissions.

## 3.2 Spatial distribution at national scale

**3.2.1 Total $CO_2$ emissions**

Since all five inventories (ODIAC, EDGAR, MEIC, CAMS, and GEMS) contain spatially explicit emission estimates for 2019, which is the latest year covered in GEMS version used in this study, we chose 2019 as the reference year for comparing the spatial patterns (Fig. 4) and the differences between the inventories using MEIC as a baseline (Fig. 5). As expected, the highest emissions are concentrated in Eastern China—especially in the North China Plain (NCP), the Beijing-

Tianjin-Hebei (BTH), the Yangtze River Delta (YRD) and the Pearl River Delta (PRD)—as hotspots of anthropogenic $CO_2$ emissions due to high population density and industrial activity (Fig. 4a-e). ODIAC shows the most intense emissions in the eastern regions, but has large spatial gaps in the west, as it relies on nighttime lighting that does not capture emissions in poorly lit areas (Fig. 4a). This approach tends to over-allocate emissions to brightly lit urban areas, while regions with limited nighttime lighting, including both sparsely populated areas and areas with high population but limited lighting, such

as Western Sichuan, Inner Mongolia, and Xinjiang, are not captured. By contrast, the spatial gaps over western China in CAMS (Fig. 4d) mainly arise from the lack of aviation emissions. CAMS accounts for transport emissions from road, off-road, and ships but omits aviation. As shown in Figure S1, EDGAR, MEIC, and GEMS capture distinct emission bands along major flight corridors over western China, whereas CAMS only shows the road transport pattern, explaining the missing emissions over western China.


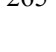

**Figure 4.** Spatial distribution of $CO_2$ emissions in 2019 at a resolution of 0.1° from ODIAC (a), EDGAR (b), MEIC (c), CAMS (d), and GEMS (e), together with the mean (f) and standard deviation (SD) (g) of the emission inventories. Sub-graph (h) shows the scatter plot illustrating the correlation between the grid-level mean emissions and the standard deviation.


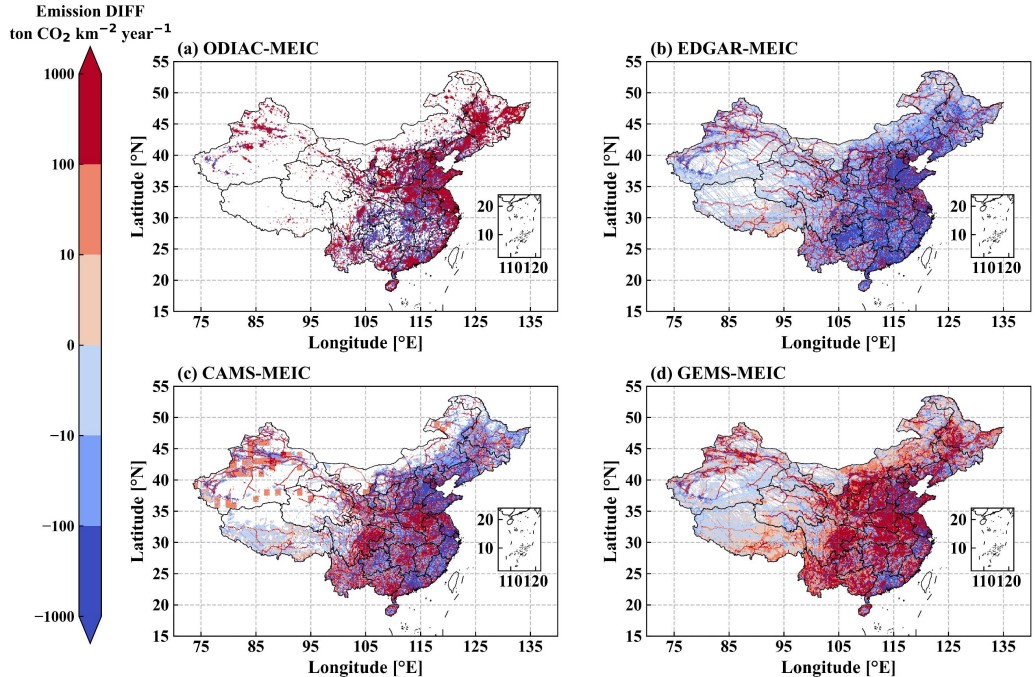

**Figure 5.** Spatial distribution of $CO_2$ emission differences in 2019 between MEIC and each of the other inventories: (a) ODIAC minus MEIC, (b) EDGAR minus MEIC, (c) CAMS minus MEIC, and (d) GEMS minus MEIC.

The SD between the five inventories (Fig. 4g) is strongly correlated with the mean of the emissions (Fig. 4f), with a slope of 0.93 and a correlation coefficient (R) of 0.95 between log-transformed estimates (Fig. 4h). This indicates that emission uncertainties are highly correlated with emission levels, and that higher uncertainties coincide with higher emissions in economic and industrial regions such as NCP, BTH, YRD, and PRD.

  To assess spatial consistency, we compared ODIAC, EDGAR, CAMS, and GEMS with MEIC as a benchmark (Fig. 5).
MEIC was chosen because it is compiled using local statistics and has been widely applied and validated in previous studies (Li et al., 2017b; Zheng et al., 2021; Che et al., 2022; Yang et al., 2025), making it a reasonable reference for comparison. Compared to MEIC, ODIAC allocates more emissions in most coastal areas and northeastern provinces (e.g., Shandong, YRD, BTH, PRD, and Northeast China), but distributes lower $CO_2$ emissions in the southwest region (e.g., Guizhou, Chongqing), where population density is relatively high but satellite nightlight signals are weak (Fig. 5a). CAMS shows an
opposite pattern, reporting lower emissions in most coastal and northeastern areas, but slightly higher values in parts of Jiangsu and Guangdong (Fig. 5c). GEMS shows slightly lower emissions in remote western areas (e.g., Xinjiang, Tibet, western Inner Mongolia) and relatively higher values in eastern provinces (Fig. 5d).

  Across the spatial domain, EDGAR generally reports lower emissions than MEIC, with negative differences prevailing throughout the region (Fig. 5b). Positive differences, which are mainly concentrated along road distribution, are much rarer
(only 39% of the number of negative difference grids). Despite this pattern, EDGAR yields a higher average grid-cell

difference from MEIC (110.60 ton $CO_2$ km$^{-2}$ year$^{-1}$) than GEMS (43.12 ton $CO_2$ km$^{-2}$ year$^{-1}$), and is only moderately lower than ODIAC (171.22 ton $CO_2$ km$^{-2}$ year$^{-1}$) and CAMS (168.80 ton $CO_2$ km$^{-2}$ year$^{-1}$). This suggests that although the positive differences between EDGAR and MEIC are spatially limited, they might be large in magnitude, potentially linking to emission hotspots such as highways or industrial clusters. We explore this further in Section 3.2.2.

### 295   3.2.2 Sectoral $CO_2$ emissions in EDGAR

To explain the higher average grid-cell emissions of EDGAR (110.60 ton $CO_2$ km$^{-2}$ year$^{-1}$ higher than MEIC in 2019) despite predominantly negative spatial differences, we analyze the discrepancies at the grid level (Fig. 6a). The cumulative sum of positive emission differences exceeds that of the negative ones when the absolute differences exceed $10^5$ ton $CO_2$ km$^{-2}$ year$^{-1}$. Although these extremes accounted for only 0.14% of the total grids, their cumulative magnitude ($1.97 \times 10^8$ ton $CO_2$ km$^{-2}$
year$^{-1}$) is 1.91 times the absolute sum of all remaining grids ($\leq 10^5$ ton $CO_2$ km$^{-2}$ year$^{-1}$, totaling $-1.03 \times 10^8$ ton $CO_2$ km$^{-2}$ year$^{-1}$). This confirms that the positive average grid-cell difference of EDGAR is caused by a small number of grids with extremely high emissions ($>10^5$ ton $CO_2$ km$^{-2}$ year$^{-1}$).

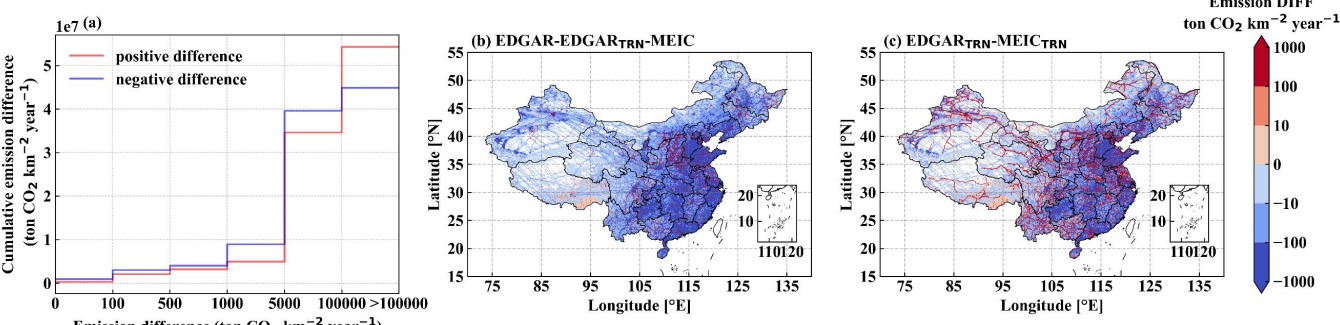

**Figure 6.** (a) Cumulative distribution of gridded emission differences (ton $CO_2$ km-2 year-1) between EDGAR and MEIC inventories. The cumulative sum for negative differences (blue line) is calculated using their absolute magnitudes and plotted against the corresponding positive values on the x-axis (i.e., 100 represents -100). The spatial distributions of the differences are shown in (b) EDGAR emissions without transport minus MEIC total emissions and (c) EDGAR transport emissions minus MEIC transport emissions.

Spatially, most of the grids with positive emission differences are shown along major road networks (Fig. 5b). When the EDGAR's transport sector is removed (Fig. 6b), the proportion of positive grids reduces drastically from 28.55% to 9.40%, confirming that the EDGAR's road transport emissions produce spatially extensive positive differences. However, the number of extreme positive emission differences ($>10^5$ ton $CO_2$ km$^{-2}$ year$^{-1}$) remains unchanged after removing transport, suggesting that these extreme differences originate from non-transport sectors. A sectoral breakdown confirms that industry
and construction contribute the most to the overall differences (1.16 Gt year$^{-1}$), followed by electricity and heat production (0.56 Gt year$^{-1}$), while residential and commercial (–0.28 Gt year$^{-1}$), and transport (–0.06 Gt year$^{-1}$) play a smaller role. Given these magnitudes, we conclude that the extremely high emitters—though few in number—are most likely from

localized industrial and power generation activities, where EDGAR may allocate emissions more aggressively to point sources than MEIC. This divergence may stem from EDGAR's use of the CARMA power plant database, while MEIC uses

CPED. Although CARMA and CPED report similar total emissions (2% difference), CPED contains approximately 1300 more small power plants (F. Liu et al., 2015; Han et al., 2020a). CARMA's sparser coverage concentrates emissions at fewer locations, thus producing EDGAR's extreme positive grid anomalies.

Despite the small total transport discrepancy ($< 0.06$ Gt year$^{-1}$) between EDGAR and MEIC, their spatial patterns differ significantly (Fig. 6c). EDGAR concentrates transport emissions along major road networks, while MEIC distributes them

more diffusely across China, which links to the different spatial allocation methods of EDGAR and MEIC. Notably, including transport emissions reduces the proportion of positive emission differences from 46.38% (non-transport only) to 28.55% (total difference). This indicates that the transport sectors of EDGAR and MEIC play a key role in the spatial pattern of positive emission differences, even though their total emissions are comparable.

### 3.3 $CO_2$ emission estimates at provincial level

### 3.3.1 Provincial estimates in CEADs

CEADs provides two forms of $CO_2$ emission estimates for provinces: the "province" series (referred to as CEADs (provinces)), which provides total emissions directly for each province, and the "sectors" series (referred to as CEADs (sectors)), which compiles fuel- and sector-specific emissions before summing them to the provincial totals. Significant discrepancies are observed between these two estimates in some provinces, with Shanxi emerging as a pronounced outlier.

After 2012, the difference in Shanxi exceeds 900 Mt year$^{-1}$, whereas in other provinces it remains below 400 Mt year$^{-1}$ (Fig. S2). To investigate this divergence, we compare both CEADs estimates with other inventories in Shanxi (Fig. 7a). The results indicate that CEADs (provinces) exceeds CEADs (sectors) after 2008, with the discrepancy growing from 167.03 Mt year$^{-1}$ in 2008 to 1167.73 Mt year$^{-1}$ in 2021. In contrast, the CEADs (sectors) closely matches the other five independent inventories (ODIAC, EDGAR, MEIC, CAMS and GEMS), with its mean emissions deviating by no more than 3.84 Mt year$^{-1}$

from the average of the five inventories. The large discrepancy between CEADs (provinces) and CEADs (sectors) mainly originates from the much higher raw coal–related emissions in CEADs (provinces) (Fig. S3), as coal is the dominant contributor to total emissions (Wei, 2022).

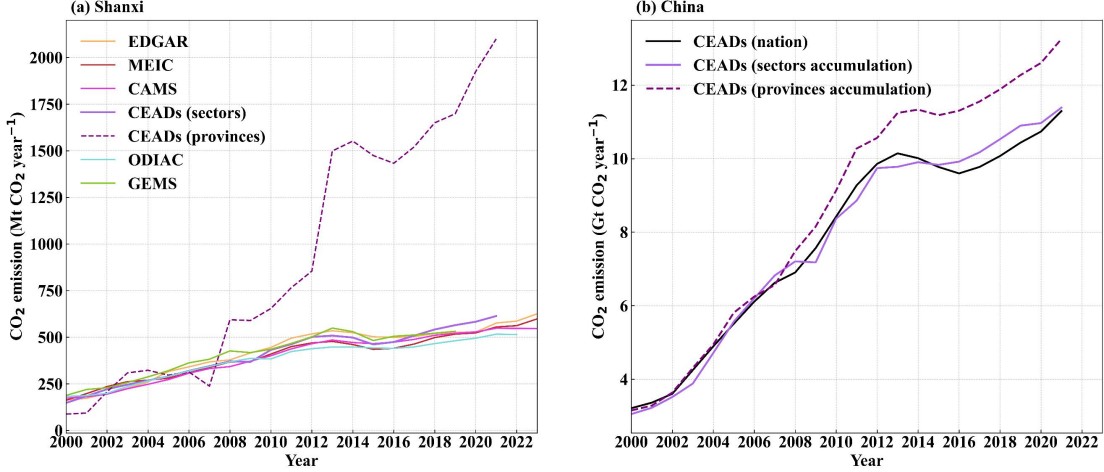


**Figure 7.** (a) Anthropogenic $CO_2$ emissions in Shanxi Province from six inventories: EDGAR, MEIC, CAMS, CEADs, ODIAC, and GEMS. CEADs provides two types of provincial-level estimates: reported provincial-level totals ("CEADs (provinces)") and aggregated sectoral emissions ("CEADs (sectors)"). Emissions from other inventories were derived by spatial aggregation of raster data. (b) Comparison between total national emissions from CEADs and the sum of provincial level emissions from CEADs (sectors) and CEADs
(nation).

At the national level, we assess both provincial datasets by aggregating their values across all provinces and comparing the results with the national total reported by CEADs (Fig. 7b). When the CEADs (sectors) are summed, the reconstructed national $CO_2$ emissions match the national CEADs values almost perfectly, showing a mean annual deviation of only 0.01 Gt

year[-1] over the period 2000-2021. In contrast, the aggregated CEADs (provinces) reports significantly higher national totals and exceeds the national CEADs emissions by an average of 0.85 Gt year[-1]. These comparisons demonstrate that the sector-based CEADs provides consistent provincial totals that are in line with both the independent inventories and the national compilation of CEADs. We therefore recommend using the CEADs (sectors) for all analyses at the national and provincial levels.

**3.3.2 Comparison of emission inventories in typical provinces**

The mean and SD of the provincial $CO_2$ emissions from 2000 to 2023 are shown in Figure S4. To investigate how inter-inventory consistency and discrepancies vary across provinces with high emissions or uncertainties, we select a subset of representative provinces for a detailed comparison. Representative provinces are identified using the SD and the mean emissions between the six emission inventories, calculated for the period 2000-2023. Each year, all provinces are ranked in

descending order based on these two metrics. The cumulative scores are calculated by summing the annual ranks over the entire 24-year period (2000-2023), reflecting each province's long-term ranking in terms of emission magnitude or SD. A lower cumulative score indicates higher mean emissions or emission uncertainties (SD). The top six provinces in each category are selected, resulting in a list of nine representative provinces (some provinces repeat in the ranking of the two

metrics): Inner Mongolia, Liaoning, Hebei, Shandong, Henan, Hubei, Shanghai, Jiangsu, and Guangdong (Table 3). In the third emissions phase (2016–2023), each of the six provinces with the highest emissions contributes more than 5.4 % of total national emissions, and together they account for almost 40 % of China's emissions. To investigate the emission patterns and cross-inventory agreement, we examine the $CO_2$ emissions of these nine representative provinces (Fig. 8).

**Table 3.** The top six provincial-level regions with the highest cumulative $CO_2$ emissions and the highest SD among the inventories (2000–2023), and $CO_2$ emission percentage of the top six provinces with the highest emissions from 2016 to 2023.

| Top six provinces by mean emissions | Cumulative rank score | $CO_2$ emission fractions (2016-2023) | Top six provinces by SD | Cumulative rank score |
|---|---|---|---|---|
| Shandong | 24 | 8.43% | Hubei | 67 |
| Jiangsu | 53 | 7.48% | Hebei | 69 |
| Hebei | 72 | 6.37% | Guangdong | 106 |
| Guangdong | 112 | 5.71% | Liaoning | 114 |
| Henan | 115 | 5.41% | Shandong | 120 |
| Inner Mongolia | 148 | 6.15% | Shanghai | 136 |

Note: Cumulative rank score refers to the sum of a province's annual rank (from highest to lowest) in terms of mean emissions or inter-inventory standard deviation (SD)

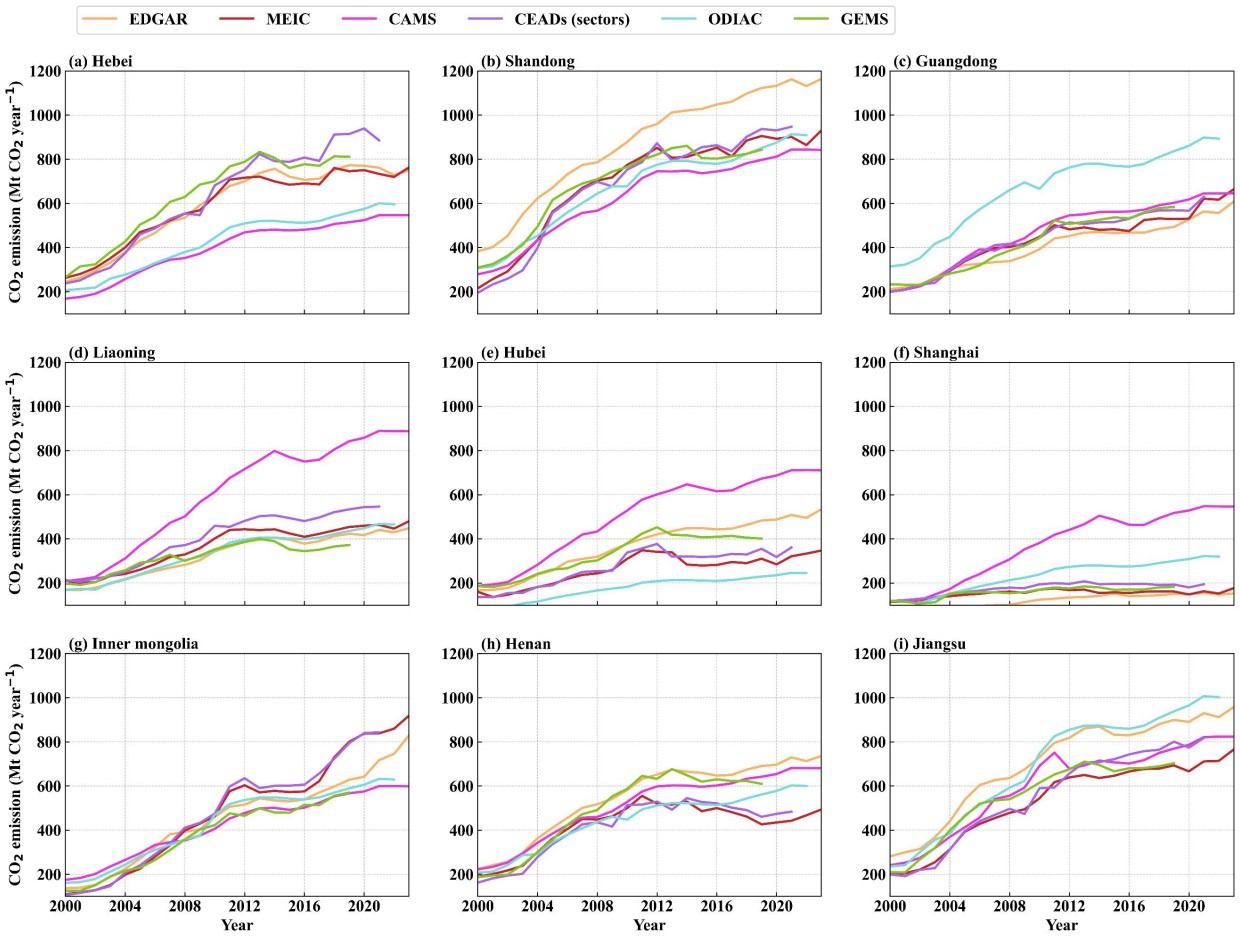

**Figure 8.** Anthropogenic $CO_2$ emissions from 2000 to 2023 for nine typical provinces: Hebei (a), Shandong (b), Guangdong (c), Liaoning (d), Hubei (e), Shanghai (f), Inner Mongolia (g), Henan (h), and Jiangsu (i). These provinces are selected based on either the highest average emissions or the highest SD among the inventories.

Among the provinces with higher emissions, Hebei, Shandong, and Guangdong rank at the top in terms of both mean emissions and SD (Table 3). In Hebei (Fig. 8a), CAMS and ODIAC report emissions averaging 416 Mt year$^{-1}$, which is 32% less than the other four inventories (618 Mt year$^{-1}$), thereby contributing significantly to the SD. In Shandong (Fig. 8b), all inventories show increased emissions, but EDGAR (873 Mt year$^{-1}$ on average) reports emissions over 30% higher than the others (670 Mt year$^{-1}$), resulting in a pronounced dispersion. Guangdong (Fig. 8c) shows a pronounced ODIAC bias, with an average of 663 Mt year$^{-1}$, over 53% higher than the average of the other five inventories (433 Mt year$^{-1}$). It is noteworthy that ODIAC significantly distributes more emissions in Jiangsu, Shanghai and Guangdong—especially in the latter two provinces. This suggests that the downscaling approach in ODIAC may overweight emissions in dense urban agglomeration (or city cluster).

Liaoning, Hubei, and Shanghai (Fig. 8d-f) are selected due to their larger inter-inventory SD. In these provinces, CAMS exceeds the mean of the five inventories by 50-90% in Liaoning, 60-110% in Hubei, and 50-230% in Shanghai, which

increases the dispersion. In Hubei, the high SD is also due to persistent dispersion across all six inventories (Fig. 8e). CAMS consistently provides the highest estimates, while ODIAC provides the lowest, making Hubei the province with the highest SD, despite average $CO_2$ emissions being only moderate.

Inner Mongolia, Henan, and Jiangsu (Fig. 8g-i) are selected for their high emissions rather than their extreme dispersion. Inner Mongolia followed the national three-stage growth pattern, with MEIC and CEADs — both China-tailored inventories — matching within 11 Mt year$^{-1}$ and even outperforming other inventories after 2016 (Fig. 8g). In Henan, domestic inventories (MEIC and CEADs) show two distinct phases: growth until 2013, followed by a decline, while the other global-based inventories (except GEMS) slowly increase after 2016 (Fig. 8h). In Jiangsu, all inventories show a two-phase trend, with rapid growth before 2013 and relative stabilization thereafter. After 2013, ODIAC and EDGAR report the highest emissions in Jiangsu, while MEIC shows the lowest trend (Fig. 8j). In the nine provinces, CEADs and MEIC estimates are largely consistent, especially in Inner Mongolia, Shandong, Henan, Hubei, and Shanghai.

Comparing the variability of emissions in the nine provinces and at the national level, the coefficient of variation (CV = SD/mean; Fig. S5) for total national emissions in China is the lowest and most stable for the period 2000-2023. In contrast, the time-averaged CV of the nine provinces with high emissions is at least 2.8 times higher than the national average (0.044). Liaoning, Hubei, and Shanghai, which show the largest SD between inventories, have even higher CVs, with values of 0.45, 0.34, and 0.26, respectively. These values exceed the national CV by a factor of 5, while Shanghai's CV exceeds the national CV by a factor of 10. This contrast emphasizes that the uncertainties at the provincial level (10-50%) are larger than the deviations at the national level (<5%), which is due to systematic biases in certain inventories and their different downscaling methods. We suggest establishing more ground-based $CO_2$ monitoring sites to verify and estimate anthropogenic $CO_2$ emissions in these provinces.

## 4 Conclusions and discussion

China's annual anthropogenic $CO_2$ total emission increases from 3.42 Gt in 2000 to 12.03 Gt in 2023. When compared with the officially reported NGHGI and the six-inventory mean, CAMS shows the smallest deviation from the NGHGI, while ODIAC agrees most closely with the multi-inventory mean. The six inventories display a broadly consistent emission trend, but their discrepancies among the inventories have widened from 0.41 Gt year$^{-1}$ to 1.63 Gt year$^{-1}$, mainly due to the highest estimates reported from EDGAR and the lowest values estimated from MEIC, especially after 2012. Our results are consistent with Zheng et al. (2025) but opposite to Han et al. (2020b), demonstrating the differences in emission versions (Our study: EDGAR2024, MEIC-global-$CO_2$ v1.0; Zheng: EDGAR v7.0, MEIC-China-$CO_2$ v1.4; Han: EDGAR v4.3.2, MEIC-China-$CO_2$ v1.3). A comparison between these versions (Fig. S6) shows that the divergence mainly arises from a downward revision in the latest MEIC dataset, which reports about 1.43 Gt year$^{-1}$ lower emissions on average over 2008–2017. In contrast, EDGAR's national totals remained nearly unchanged across versions, with differences within 0.001 Gt

year$^{-1}$ during 2000-2012. These results highlight the significant impact of inventory version updates on comparative emission analyses.

The six inventories in this study agree on three emission phases: a rapid increase of $0.56 \pm 0.013$ Gt year$^{-1}$ (2000–2013), a near-stagnation phase of $-0.07 \pm 0.022$ Gt year$^{-1}$ under the 12th Five-Year Plan and air clean policy (2013–2016), and a renewed growth of $0.30 \pm 0.016$ Gt year$^{-1}$ (2016–2023), mainly related to infrastructure-driven energy demand and coal use recovery following 2016. In terms of emission sectors, emissions are dominated by electricity and heat production, industry and construction (together accounting for 78% of total emissions). The former source overtook the latter as the largest source after 2017, reflecting changes in China's energy structure.

In spatial terms, the higher emissions strongly corresponded with the higher uncertainty (reference 2019: R = 0.95, P< 0.01). Eastern regions, particularly the BTH, YRD, and PRD city clusters, had both the highest emissions and the largest SD. This pattern confirms the finding of Wang et al. (2013) that areas with high emission level have the largest uncertainties. Different allocation methods are the main reason for the spatial discrepancies between the inventories. The ODIAC nightlight proxy distributes more emissions in urban areas and fewer emissions in the western regions. EDGAR, which is based on the CARMA database, concentrated power plant emissions on fewer grids, resulting in extreme anomalies where the difference (EDGAR-MEIC) exceeds $10^5$ ton $CO_2$ km$^{-2}$ year$^{-1}$. These high-value grids underscore the importance of cross-inventory comparisons when using EDGAR to analyze the spatial distribution of industry sector or power plant emissions in China. In contrast, MEIC uses the more detailed CPED and distributes similar total $CO_2$ emissions (difference within 2% of CARMA) across a larger number of power plants (Liu et al., 2015). The overall spatial grid-based difference between EDGAR and MEIC is dominated by negative values (71.45% of grids), due to the different allocation methods for the transport sector. EDGAR allocates emissions along major roads, while MEIC uses a more diffuse distribution. Despite a minimal overall difference in the sector of transport (< 0.06 Gt), the spatial mismatch was substantial, with 70.37% of transport-related grid differences being negative, due to the different disaggregation methods: OpenStreetMap and OpenRailwayMap in EDGAR versus CDRM in MEIC.

At the provincial level, CEADs data show critical inconsistencies: its provincial sectoral emissions are consistent with the multi-inventory means, but the provincial series reports lower emissions in Shanxi by more than 127% (approximately 500 Mt year$^{-1}$). We therefore recommend sector-based CEADs for province-level analyses. The uncertainty in the province scale is significantly higher than the national scale. For example, the coefficient of variation (CV) of Shanghai (0.45) is ten times higher than the national CV (0.044). The pronouncedly higher emissions in the coastal megacities (e.g., Shanghai, Jiangsu, and Guangdong) by ODIAC and the abnormal increase in CAMS by 50-230% in Liaoning, Hubei, and Shanghai exacerbate this divergence. Despite these inconsistencies, CEADs and MEIC exhibit broadly consistent estimates across nine provinces, especially in Inner Mongolia, Shandong, Henan, Hubei, and Shanghai.

In summary, this study extends previous work by identifying a three-phase trend in China's anthropogenic $CO_2$ emissions from 2000 to 2023 and quantifying the emission uncertainties (1σ) at both national and provincial levels. At the national level, CAMS shows the closest agreement with the government-reported NGHGI, while ODIAC aligns best with the multi-

inventory mean over the study period. At the provincial level, the Chinese local inventories, CEADs and MEIC, provide the most consistent estimates for regional studies. Differences in spatial proxies significantly affect the spatial distribution of sectoral emissions, as shown by the contrasting transport emission patterns in EDGAR and MEIC. We also clarify the appropriate use of CEADs for provincial analyses. Our results further underscore the importance of improving the consistency of regional inventories to provide a stronger scientific basis for China's emission mitigation and carbon neutrality policies.

Overall, reliable emissions quantification requires scale-appropriate inventories (e.g., the sectoral CEADs emissions versus the province-based CEADs emissions), improved spatial proxies (e.g., CPED vs. CARMA), and ensemble approaches to mitigate biases, especially in the carbon-intensive eastern regions. It should be noted that this study lacks an observational benchmark to assess these inventories. Future efforts should incorporate direct flux measurements or top-down emissions derived from inversion modeling, in combination with $CO_2$ mole fraction observations, to compare and constrain bottom-up inventories at both regional and national scales.

**Data availability**

The emission inventories datasets are publicly available: ODIAC (https://db.cger.nies.go.jp/dataset/ODIAC/DL_odiac2023.html), EDGAR (https://edgar.jrc.ec.europa.eu/dataset_ghg2024), MEC (http://meicmodel.org.cn/?page_id=2341), CAMS (https://eccad.sedoo.fr/#/metadata/479), GEMS (https://gems.pku.edu.cn/home) and CEADs (https://www.ceads.net.cn/).

**Author contribution**

HY, KW, and MZ designed the study. HY evaluated the data and wrote the paper with the inputs from KW and MZ. HS and GJ-M provided valuable suggestions to improve the manuscript. All authors read and provided comments on the paper.

**Competing interests**

The authors declare that they have no conflict of interest.

**Acknowledgements**

This study is supported by the National Key Research and Development Program of China (No. 2023YFB3907500, 2023YFB3907505), the Basic Research Project of the Institute of Atmospheric Physics, Chinese Academy of Sciences (E468131801) and the State Key Laboratory of Atmospheric Environment end Extreme Meteorology (NO. 2024QN04). We gratefully acknowledge the data providers of the emission inventories used in this study: the ODIAC dataset developed by

the Global Carbon Project Center for Global Environmental Research (CGER) at the National Institute for Environmental Studies (NIES), Japan, led by Dr. Tomohiro Oda; the EDGAR dataset maintained and publicly released by the Joint Research Centre (JRC) of the European Commission; the MEIC inventory developed by Tsinghua University under the direction of Dr. Qiang Zhang; the CAMS inventory jointly developed by the Copernicus Atmosphere Monitoring Service
(CAMS) at the European Centre for Medium-Range Weather Forecasts (ECMWF) on behalf of the European Commission, with contributions from Dr. Antonin Soulie; the GEMS inventory developed by a joint team from Peking University and Southern University of Science and Technology; and the CEADs inventory, also developed by Tsinghua University, with contributions from Dr. Dabo Guan.

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

# 7 Supplementary materials

**Table S1.** Definition of sub-sectors included in the four main sectors (electricity and heat production, industries and construction, transport, and residential and commercial) of the MEIC inventory.

| Sector in MEIC-global-$CO_2$ | Definition |
| --- | --- |
| Electricity and heat production | Power generation |
| | Heat (auto producer) |
| | Heat (public) |
| Industries and construction | Coal mines |
| | Oil and gas extraction |
| | Blast furnaces |
| | Gas works |
| | Gasification plants for biogases |
| | Coke ovens |
| | Patent fuel plants |
| | BKB/peat briquette plants |
| | Oil refineries |
| | Coal liquefaction plants |
| | Liquefaction (LNG) / regasification plants |
| | Gas-to-liquids (GTL) plants |

|  |  |
|---|---|
|  | Own use in electricity, CHP and heat plants |
|  | Charcoal production plants |
|  | Non-specified transformation industries |
|  | Iron and steel |
|  | Non-ferrous metals |
|  | Chemicals |
|  | Pulp and paper |
|  | Food and tobacco |
|  | Cement |
|  | Other non-metallic minerals |
|  | Transport equipment |
|  | Machinery |
|  | Mining and quarrying |
|  | Wood products |
|  | Construction |
|  | Textile and leather |
|  | Other non-specified industries |
|  | Process emissions in cement industry |
| Transport | International aviation[1] |
|  | Domestic aviation |
|  | Rail |
|  | International navigation[1] |
|  | Domestic navigation |
|  | Pipeline transport |
|  | Other non-specified transport |
|  | Agriculture and forestry |
|  | Fishing |
|  | Cars |
|  | Light duty trucks |
|  | Buses |
|  | Heavy duty trucks |
|  | Motorcycles |
|  | Other fleet totals |

| | Commercial and institutional |
| Residential and commercial | Residential (rural) |
| | Residential (urban) |
| | Non-specified sectors |


**Table S2.** Definition of sub-sectors in the four main sectors of EDGAR, CAMS, CEADs and GEMS.

| Emission inventory | Sector | Definition |
|---|---|---|
| EDGAR | Electricity and heat production | Main Activity Electricity and Heat Production |
| | Industries and construction | Petroleum Refining - Manufacture of Solid Fuels and Other Energy Industries |
| | | Manufacturing Industries and Construction |
| | | Solid Fuels |
| | | Oil and Natural Gas |
| | | Cement production |
| | | Lime production |
| | | Glass Production |
| | | Other Process Uses of Carbonates |
| | | Chemical Industry |
| | | Metal Industry |
| | | Non-Energy Products from Fuels and Solvent Use |
| | | Liming |
| | | Urea application |
| | | Incineration and Open Burning of Waste |
| | | Fossil fuel fires |
| | Transport | Civil Aviation |
| | | Road Transportation no resuspension |
| | | Railways |
| | | Water-borne Navigation |
| | | Other Transportation |

| | | Residential and other sectors |
|---|---|---|
| | Residential and commercial | Non-Specified |
| CAMS | Electricity and heat production | Power generation |
| | | Fugitives |
| | | Industrial process |
| | Industries and construction | Refineries |
| | | Solid waste and waste water |
| | | Solvents |
| | | Off Road transportation -China |
| | Transport | Road transportation |
| | | Ships |
| | Residential and commercial | Commercial |
| | | Residential |
| CEADs | Electricity and heat production | Production and Supply of Electric Power, Steam and Hot Water |
| | Industries and construction | Coal Mining and Dressing |
| | | Petroleum and Natural Gas Extraction |
| | | Ferrous Metals Mining and Dressing |
| | | Nonferrous Metals Mining and Dressing |
| | | Nonmetal Minerals Mining and Dressing |
| | | Other Minerals Mining and Dressing |
| | | Logging and Transport of Wood and Bamboo |
| | | Food Processing |
| | | Food Production |

Beverage Production

Tobacco Processing

Textile Industry

Garments and Other Fiber Products

Leather, Furs, Down and Related Products

Timber Processing, Bamboo, Cane, Palm Fiber & Straw Products

Furniture Manufacturing

Papermaking and Paper Products

Printing and Record Medium Reproduction

Petroleum Processing and Coking

Raw Chemical Materials and Chemical Products

Medical and Pharmaceutical Products

Chemical Fiber

Rubber Products

Plastic Products

Nonmetal Mineral Products

Smelting and Pressing of Ferrous Metals

Smelting and Pressing of Nonferrous Metals

Metal Products

Ordinary Machinery

Equipment for Special Purposes

| | | |
|---|---|---|
| | | Transportation Equipment |
| | | Electric Equipment and Machinery |
| | | Electronic and Telecommunications Equipment |
| | | Instruments, Meters, Cultural and Office Machinery |
| | | Other Manufacturing Industry |
| | | Scrap and waste |
| | | Production and Supply of Gas |
| | | Production and Supply of Tap Water |
| | | Construction |
| | Transport | Farming, Forestry, Animal Husbandry, Fishery and Water Conservancy |
| | | Transportation, Storage, Post and Telecommunication Services |
| | Residential and commercial | Cultural, Educational and Sports Articles |
| | | Wholesale, Retail Trade and Catering Services |
| | | Others |
| | | Urban |
| | | Rural |
| GEMS | Electricity and heat production | Power Generation |
| | Industries and construction | Industrial Process |
| | | Industrial Combustion |
| | Transport | Transportation |
| | | Agriculture |
| | Residential and commercial | Commercial |

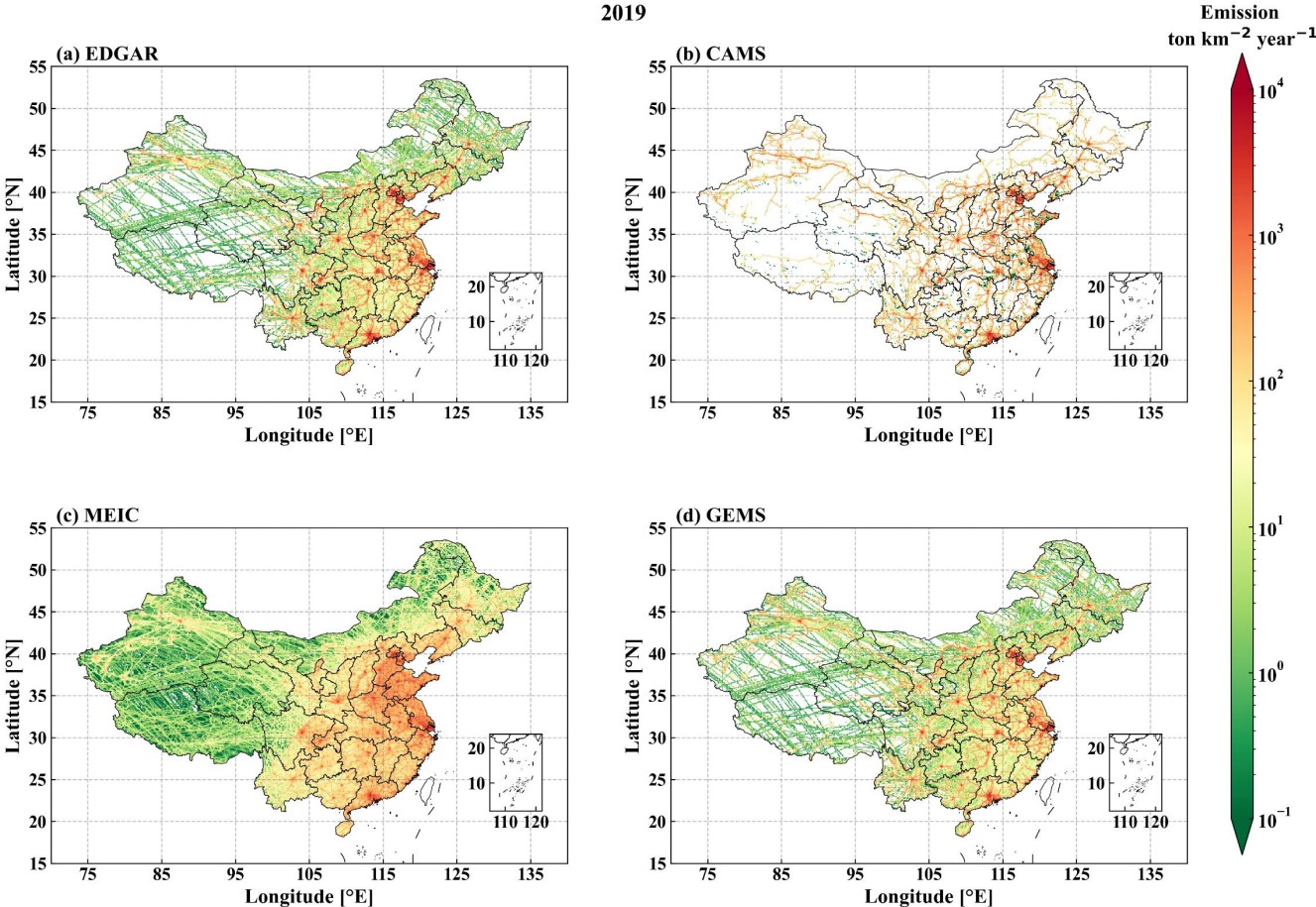

**Figure S1.** Spatial distribution of $CO_2$ emissions from transport sector in 2019 across four inventories (EDGAR, CAMS, MEIC, and GEMS).

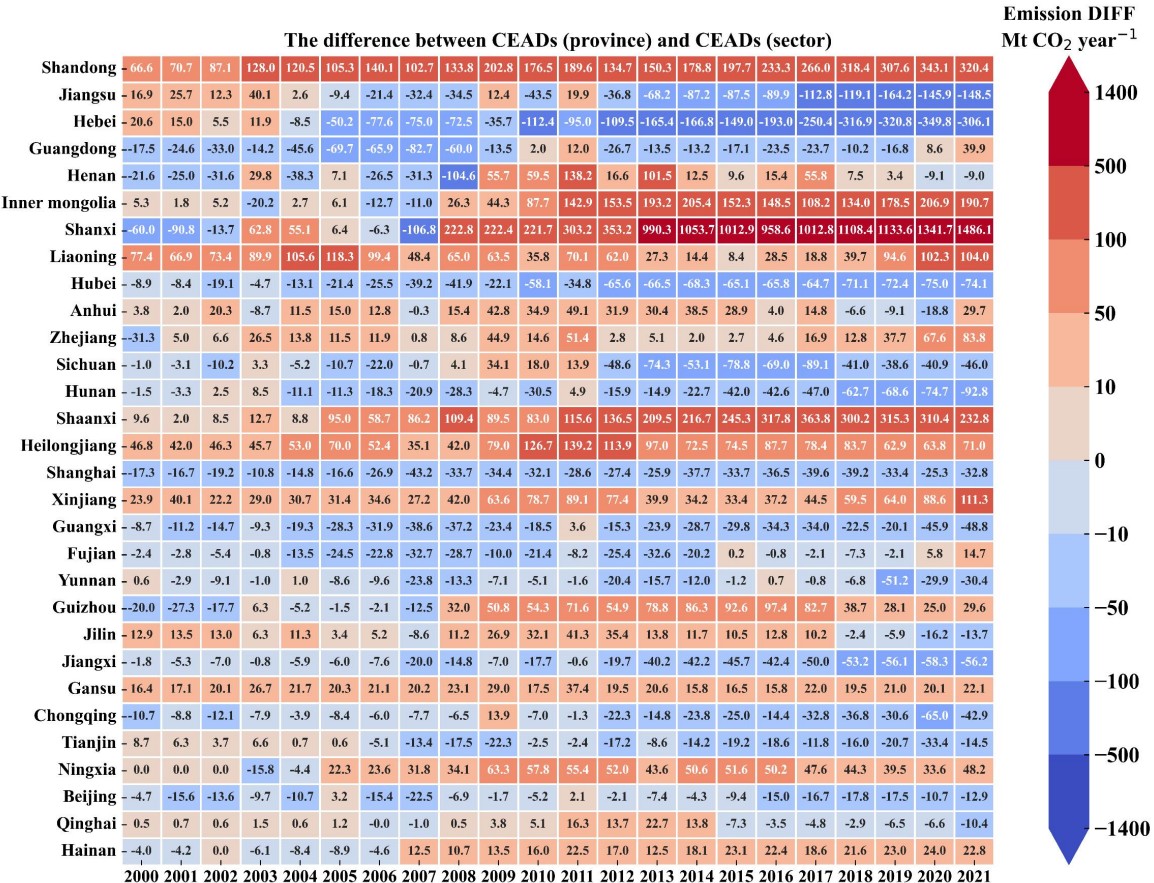

**Figure S2.** Heatmap of the annual $CO_2$ emission differences between CEADs (province) and CEADs (sector) for 30 Chinese provinces provided by CEADs during 2000–2021. Provinces are ordered by total emissions from highest to lowest.

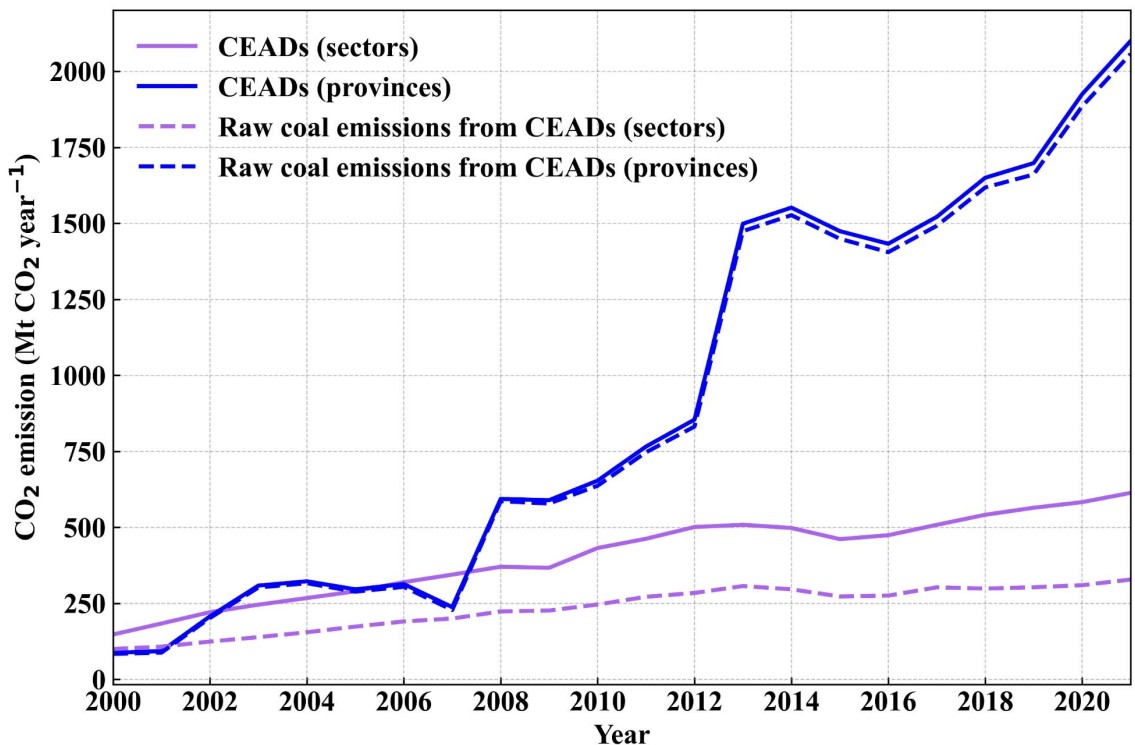

**Figure S3.** Comparison of total $CO_2$ emissions and raw coal–related $CO_2$ emissions in Shanxi from CEADs (sectors) and CEADs (provinces) during 2000–2020. Solid lines represent total emissions, while dashed lines indicate emissions from raw coal combustion.

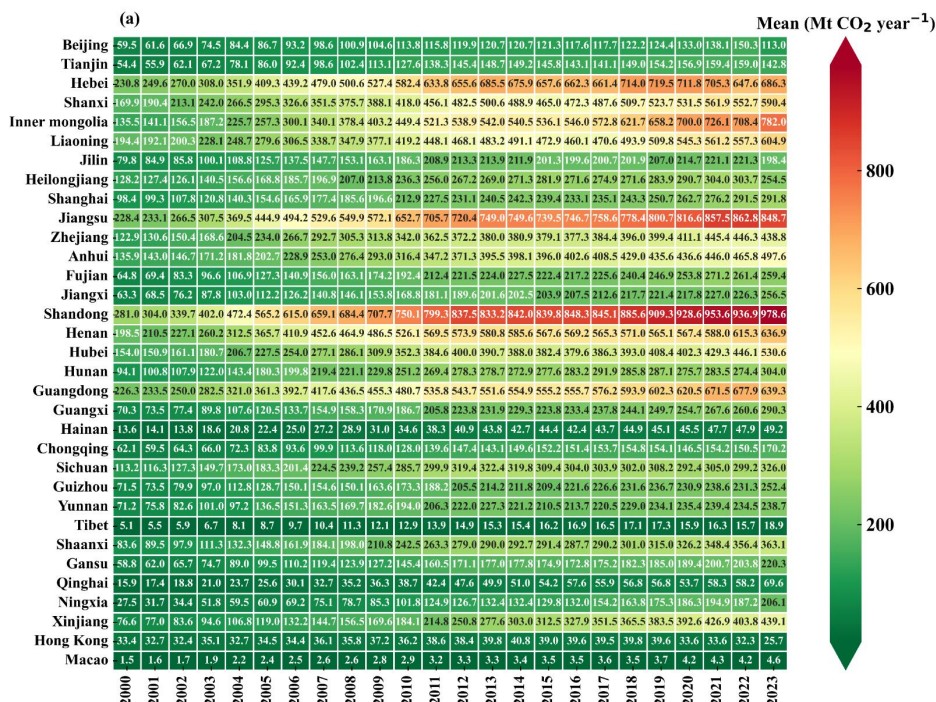

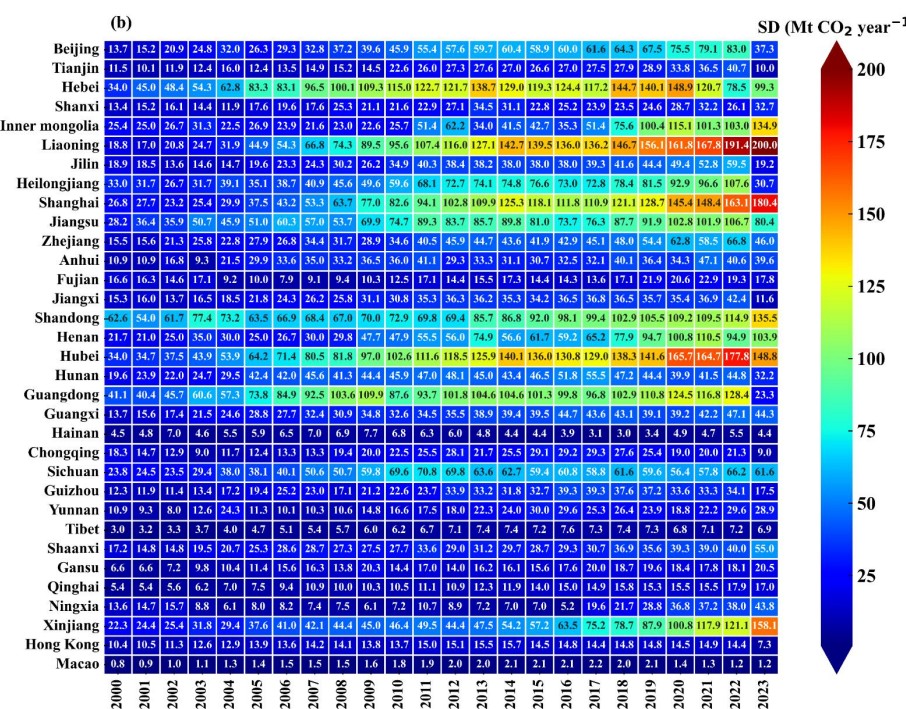

**Figure S4.** Heatmaps of provincial mean CO2 emissions (a) and SD (b) of six emission inventories for the period 2000 to 2023.


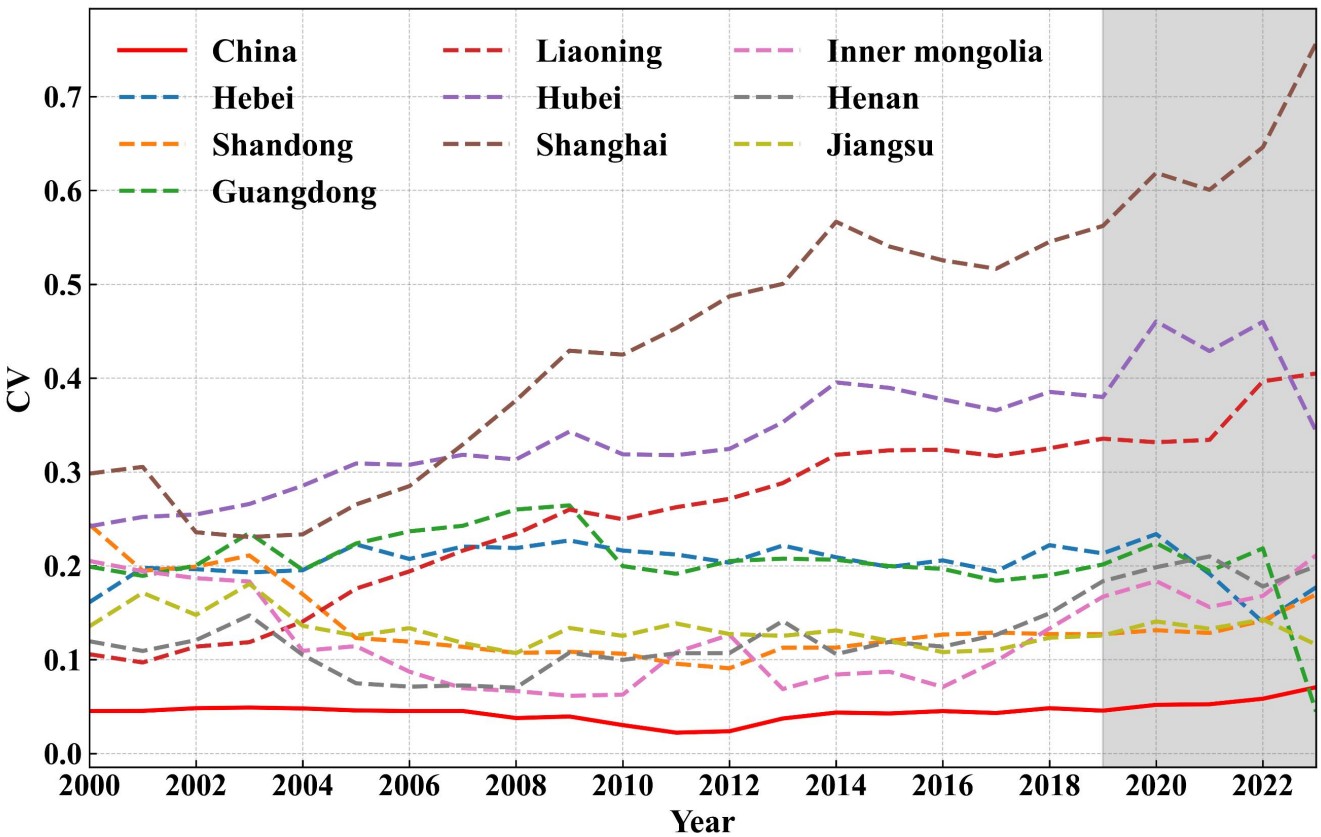

**Figure S5.** Coefficient of variation (CV) of emissions at national level and for nine typical provinces during 2000-2023. The shaded area
represents the period after 2019, when the number of available emission inventories began to decrease (GEMS ended in 2019, CEADs in
2021, and ODIAC in 2022).

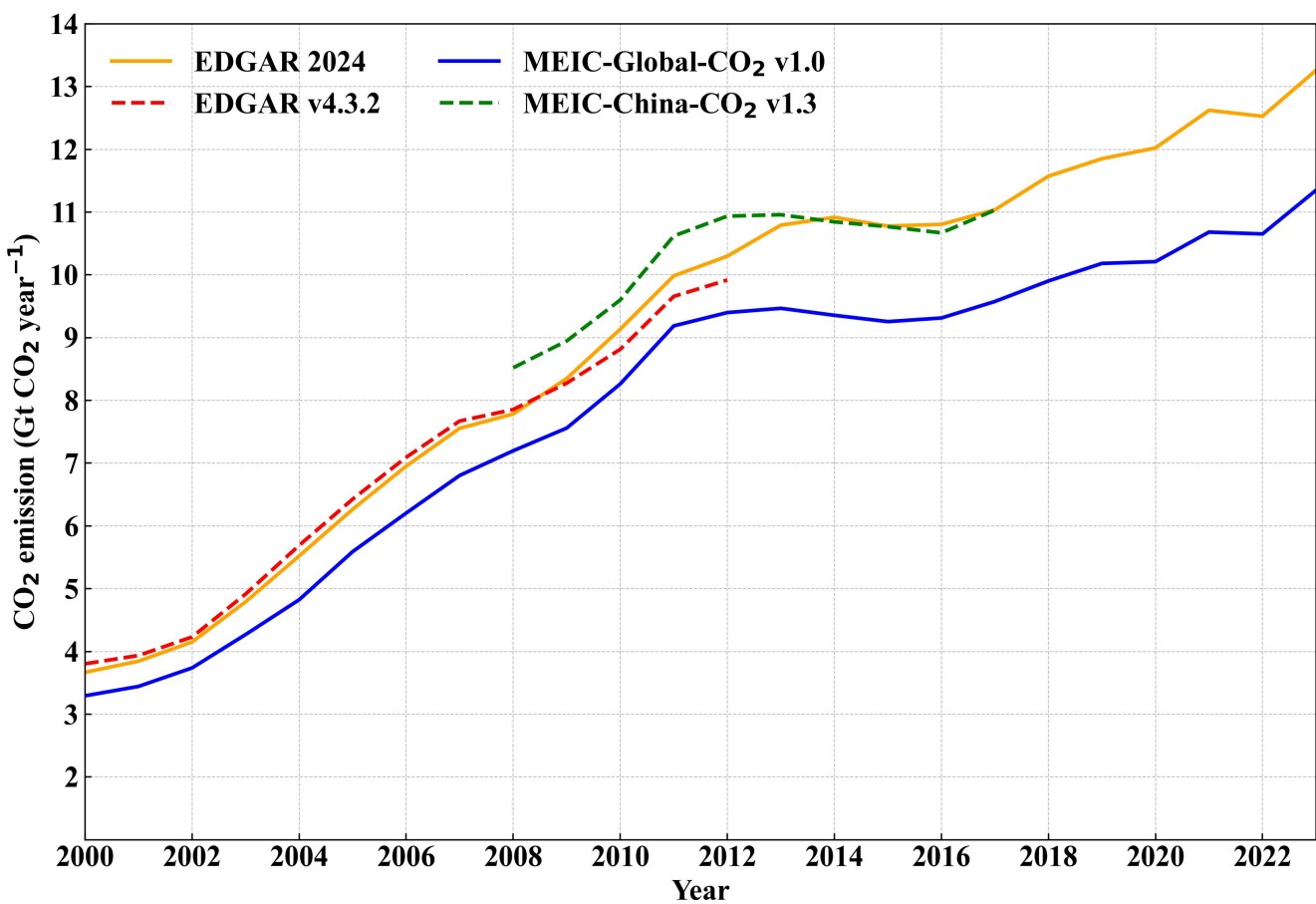

**Figure S6.** Comparison of national $CO_2$ emissions from different versions of the EDGAR and MEIC inventories. The older versions (EDGAR v4.3.2 and MEIC-China-$CO_2$ v1.3) used in Han et al. (2020b) are compared with the updated versions (EDGAR 2024 and MEIC-Global-$CO_2$ v1.0) used in this study.
