# Peer review of "A Comparative Analysis of China's Anthropogenic CO2 Emissions (2000–2023): Insights from Six Bottom-Up Inventories and Uncertainty Assessment"

_EGUsphere, 2025_

## Referee Comment (RC1)

This manuscript offers a robust comparative analysis of six $CO_2$ emission inventories for China, integrating both local and global datasets. A key strength is its detailed assessment of spatial and temporal uncertainties, an often overlooked but policy-relevant aspect. The study contributes meaningfully by highlighting inventory discrepancies and emphasizing the importance of uncertainty assessments in emission reporting. However, I have the following specific comments that require clarification and revision before the manuscript can be considered for publication.

General comments

The manuscript is clearly written and well structured, with a logical flow that facilitates understanding of the main objectives and findings.

However, it is not entirely clear whether the emission inventories selected for analysis are the only relevant options available, or what criteria guided their selection. Since the manuscript references other inventories that were ultimately not included in the comparison, it would strengthen the study to provide a clearer rationale for the choices made.

The relevance of the topic is evident, especially in light of China's pivotal role in global emissions and its commitments under the Paris Agreement. Still, the manuscript would benefit from a more explicit explanation of why comparing the latest versions of these inventories is particularly important. A clearer articulation of what distinguishes this study from previous work (beyond simply the version updates) would improve accessibility, especially for readers less familiar with the topic.

The discussion of differences between inventories and their associated uncertainties is engaging and informative. However, a clear take-home message is lacking, particularly regarding which inventories may be considered more reliable or fit for specific purposes. While it is understandable that definitive recommendations may be difficult, the current conclusions are limited, with the mainly strong guidance being to avoid the provincial CEADs inventory. Offering more concrete insights or practical recommendations, especially in the context of supporting policymaking, would significantly strengthen the manuscript.

Specific Comments

Line 35: To highlight China's role in global emissions, please include the percentage of China's anthropogenic emissions relative to global totals.

Line 44: The CAMS inventory should be included in this overview for completeness.

Line 48: Are there specific reasons for not including CHRED in the analysis? Please clarify.

Line 80: Consider introducing the CAMS inventory definition earlier in this section alongside the others, for consistency.

Line 80: MEIC is initially described (line 47) as a China-specific inventory, but here it is treated as a global inventory. This inconsistency may confuse readers, particularly since

line 116 clarifies that the global version of MEIC is used. Please harmonize these descriptions.

Line 122: The mention of the number of species covered by CAMS is not relevant here, as the analysis focuses on a single species. Also, this level of detail is not provided for the other inventories.

Line 198: Do you have any hypotheses as to why GEMS diverges from the trends observed in other inventories, especially in the residential and commercial sectors?

Table 1: Time Resolution (GEMS column): Please change "Annually" to "Annual" to align with the other entries.

Table 1: Data Source row: Since the "last accessed" date is the same for all inventories, consider moving this note to a table footnote (e.g., marked with an asterisk) to streamline the table.

Figure 3: The growth in electricity and heat production in CAMS appears to stabilize, unlike in other inventories where growth continues. Given CAMS is based on EDGAR, a similar trend would be expected. Could this discrepancy be linked to the use of CAMS-Tempo profiles?

Line 212: It is unclear why MEIC is used as a benchmark for comparison. Please add a brief explanation of this choice.

Figure 5c: What accounts for the squared patches in this figure? A brief explanation in the caption or main text would help readers interpret the results.

Figures 4 & 5: In Figure 4, MEIC shows notable emissions over western China (green shading), while ODIAC does not. This difference should manifest as strong negative values (blue) in Figure 5, yet much of this area appears blank, which I assume represents NaN values. Did you apply any filtering? Please clarify.

Line 241: For clarity, please consider rephrasing this sentence, here is a suggestion:

"Across the spatial domain, EDGAR generally reports lower emissions than MEIC, with negative differences prevailing throughout the region."

Line 287: Could the discrepancy in Shanxi be attributed to a specific sector? A sectoral analysis, as presented in the previous section, would be valuable here.

Line 290: Could you comment on the provincial comparison of the two CEADs estimates beyond Shanxi? Do any provinces show consistent agreement between the two datasets, and are these primarily low-emission regions? A colored map showing the differences between the two CEADs estimates by province could be a useful addition

---

## Referee Comment (RC3)

General comments:

This manuscript analyzes and compares six bottom-up inventories and assesses their uncertainty. This work compares different inventories from international and domestic teams, and it will be useful to the global stocktake and accurately assess China's $CO_2$ emissions. The topic is interesting and meaningful, but many statements and explanations in the manuscripts are not rigorous enough. I suggest more modifications and improvements before acceptance.

Special comments:
1   Is it reasonable to use the mean and SD to assess the uncertainty of these emission inventories?
2   Activity data and emission factors are the two important factors that influence the emission inventory. I also suggest adding this important information to Table 1, although point, line, and area source proxies are listed.
3   Chinese government also reports national greenhouse gas emissions to the UNFCCC. I think it is better to compare the national $CO_2$ emissions between government-reported data and the six bottom-up inventory data mentioned in this study.
4   Line 168-174. Many studies report that China's emissions peaked in 2013 or 2014, so the first phase is better set as 2000-2013 or 2014. Also, the second phase is mainly due to the air control policy, besides the adjustment of energy and industrial structure.
5   Line 180-185. Although the global stocktake is held every five years, the stocktake assesses the achievement of NDCs of each country. Also, the baseline year of the Chinese 2020 and 2030 carbon reduction targets is 2005. I suggest the authors rewrite these sentences.
6   Figure 4. Point and line sources of CAMS originated from EDGAR (Table 1). Why is the line source information lost in Figure 4d, especially in western China? Furthermore, the map of means (Figure 4f), most of the line and area information was lost.
7   Figure 5c. Why are there some squares with high values in the west and northeast China?
8   Figure 7. Why is the CEADs province data nearly ten times higher than other inventories in Shanxi Province?
9   Figure 8. EAGAR and MEIC are the highest and lowest inventories for national $CO_2$ emissions, but these values varied at the provincial level. What are the key factors that affected these results? For example, CAMS had the highest values in Liaoning, Hubei provinces and Shanghai but the lowest in Hebei and Shandong provinces.
10  Figures S1&S2, Why do SD and CV for Hubei and Guangdong decrease sharply in 2023?
11  Table 1. Why can CAMS report the data in 2026 when it was published in 2023?
12  Line 104. What does "BP plc" mean?

13   Table S1. Please add a footnote for "1" mentioned in the transport sector.

---

## Referee Comment (RC4)

**Review of Yang et al. (2025)**

The manuscript by Yang et al. (2025) addresses an important issue in carbon emission reporting, namely the comparison of six different bottom-up inventories using China as a case study. Accurate quantification of $CO_2$ emissions is critical for developing effective mitigation policies. The authors' approach of including three global inventories and three local inventories makes the comparison meaningful and comprehensive. The manuscript is clearly written and was enjoyable to read. I have the following specific comments that require clarification before the manuscript can be considered for publication

**Specific Comments:**

**Introduction section**

I found the introduction engaging, but a few aspects could be elaborated further:

i. Please clarify why China was selected as the case study. Is it solely because China is the world's second largest emitter of $CO_2$, or also because it provides a unique combination of global and local inventories suitable for comparison? Additionally, given that many similar studies have already been conducted for China, does this choice facilitate comparison with existing literature? Please specify.

ii. The authors have summarized previous studies from China that compared a few inventories. What is the novelty of the present work? Is the use of updated versions of inventories the only advancement, or are there other new aspects? Please state this explicitly.

**Result Section**

**Section 3.1**

The authors state that differences among the emission inventories become more pronounced after 2012 and continue to diverge in recent years. However, the manuscript does not provide an explanation for this trend. It would greatly benefit the reader if the authors elaborated on the possible reasons for this divergence—for example, changes in activity data sources, revisions in statistical reporting, or methodological updates within specific inventories. Such context is essential to help readers better understand the evolution of Chinese emissions estimates over time.

**Conclusion section**

The conclusion could be strengthened by addressing the following points:

What is the main take-home message from this study?

Which inventory performs better overall for China?

Are certain inventories more reliable in high-emission regions versus low-emission regions?

Currently, these questions remain unanswered. I think including these aspects will be helpful for readers, providing them with clearer guidance and enhancing the practical value of the study.

**Recommendation:** This manuscript has the merit and it presents valuable data. However, it requires above minor revisions to be addressed before considered for the publication in Atmospheric Chemistry and Physics journal.

---

## Author Comment (AC1)

**RC5**

The manuscript by Yang et al. 2025 provides a comparative analysis of China's anthropogenic CO2 emissions over the period 2000–2023, based on six widely used bottom-up inventories. The topic is highly relevant given the importance of accurate CO2 accounting for climate change mitigation and policy verification. The dataset selection is comprehensive, and the study offers valuable insights into temporal trends, sectoral contributions, and spatial differences across inventories. Overall, the manuscript is well written and scientifically sound.

**Comments:**

1. Although comparing multiple inventories is valuable, similar studies have been conducted for China in the past. The authors should make it clearer what distinguishes this work. Does the novelty lie mainly in the inclusion of the most recent versions of the inventories? Or is it the extension to 2023 and more detailed assessment?

Response: We appreciate the reviewer's valuable comment. To clarify the novelty of our work, we have added a short paragraph in the Introduction and a more detailed paragraph in the Conclusion explicitly outlining the main advancements compared with previous studies. Specifically, this study (1) extends the temporal coverage to 2000–2023 and identifies three distinct emission phases reflecting changes in energy policy and structure; (2) evaluates internal inconsistencies within CEADs and recommends using CEADs (sectors) for provincial analyses; (3) reveals significant sectoral spatial allocation differences, particularly between EDGAR and MEIC in the transport sector; (4) quantifies scale-dependent uncertainties, showing that provincial uncertainty (CV) is two to ten times higher than national uncertainty; and (5) demonstrates that CEADs and MEIC yield consistent estimates across nine representative provinces. At the national scale, CAMS exhibits the smallest deviation from the National Greenhouse Gas Inventory (NGHGI), while ODIAC aligns most closely with the six-inventory mean during the study period. These clarifications have been added to Section 1 to highlight the study's novelty and rationale for using the latest inventory versions, and to Section 4 to summarize the new insights contributions.

**Revision:**

- (1) **Section 1, paragraph 4:** "Moreover, emission inventories are continuously updated to incorporate improved inputs (e.g., activity data, EFs, and refined methodology). Therefore, it is crucial to use the latest versions of the various inventories to capture these methodological updates and better understand the most recent patterns of China's anthropogenic CO2 emissions."
- (2) **Section 1, paragraph 5:** "... Compared with previous studies (Han et al., 2020b; Zheng et al., 2025), we extend the temporal coverage to 2000-2023, enabling a more current and consistent assessment of recent emission trends, inter-inventory discrepancies, and scale-dependent uncertainties across China."
- (3) **Section 4, paragraph 5:** "In summary, this study extends previous work by identifying a three-phase trend in China's anthropogenic CO2 emissions from 2000 to 2023 and quantifying the emission uncertainties (1 $\sigma$ ) at both national and provincial levels. At the national level, CAMS

shows the closest agreement with the government-reported NGHGI, while ODIAC aligns best with the multi-inventory mean over the study period. At the provincial level, the Chinese local inventories, CEADs and MEIC, provide the most consistent estimates for regional studies. Differences in spatial proxies significantly affect the spatial distribution of sectoral emissions, as shown by the contrasting transport emission patterns in EDGAR and MEIC. We also clarify the appropriate use of CEADs for provincial analyses. Our results further underscore the importance of improving the consistency of regional inventories to provide a stronger scientific basis for China's emission mitigation and carbon neutrality policies."

2. The conclusion is very long and contains many technical details. The authors discuss discrepancies between the inventories and ultimately recommend one, the CEADs for sectoral data, but what is the direct impact of this on emissions control in China? The text could end with a strong recommendation to improve regional inventories aimed at supporting mitigation policies.

Response: We thank the reviewer for this valuable suggestion. We agree that highlighting the policy relevance of our findings would strengthen the conclusion. While our study focuses on technical consistency across inventories, the results have clear implications for emissions management in China. We have therefore revised the conclusion to emphasize that improving the accuracy and consistency of regional inventories is essential for tracking progress toward China's dual carbon targets and for supporting evidence-based mitigation policies. Furthermore, to ensure the reliability of inventories, we suggest expanding both ground-based and satellite observations to enable comprehensive independent validation. Specifically, CO2 flux measurements can be directly compared with bottom-up estimates, while atmospheric CO2 mole fraction measurements, when integrated with inversion models, yield top-down emission estimates. These top-down results can then be systematically compared with the bottom-up inventories to identify the discrepancies across regional and national scales. We have revised Section 4 to clarify this point.

**Revision:**

- (1) **Section 4, paragraph 5:** "... Our results further underscore the importance of improving the consistency of regional inventories to provide a stronger scientific basis for China's emission mitigation and carbon neutrality policies."
- (2) **Section 4, paragraph 6:** "Overall, reliable emissions quantification requires scale-appropriate inventories (e.g., the sectoral CEADs emissions versus the province-based CEADs emissions), improved spatial proxies (e.g., CPED vs. CARMA), and ensemble approaches to mitigate biases, especially in the carbon-intensive eastern regions. It should be noted that this study lacks an observational benchmark to assess these inventories. Future efforts should incorporate direct flux measurements or top-down emissions derived from inversion modeling, in combination with CO2 mole fraction observations, to compare and constrain bottom-up inventories at both regional and national scales."

**Specific comments:**

**1. Line 183:** What is the reason for this stable period between 2012 and 2017?

**Response:** We thank the reviewer for the question. It is worth noting that China's CO2 emissions are estimated to have peaked around 2013 according to previous studies, which is why we defined 2000–2013 as the first phase and 2013–2016 as the second in our analysis. The 2013–2016 period represents a short-term stabilization of emissions mainly driven by the adjustment of China's energy structure and industrial upgrades under the 12th Five-Year Plan, together with the implementation of national air pollution control policies since 2013 (Han et al., 2020b; Shi et al., 2022; Zheng et al., 2025). The corresponding text and linear regression statistics in the first and second emission phase has been revised for clarity.

**Revision:**

- (1) **Abstract:** "...The national total  $CO_2$  emissions increase from 3.43 (3.21–3.63) Gt year-1 in 2000 to 12.03 (11.35–12.98) Gt year-1 in 2023, with three growth periods: rapid growth (2000–2013, 0.56±0.013 Gt year-1), near-stagnation (2013–2016, -0.07±0.022 Gt year-1), ..."
- (2) **Section 3.1, paragraph 3:** "The increase in  $CO_2$  emissions shows three different phases (Fig. 1, Table 2). The first phase ( $\underline{2000-2013}$ ) shows the most rapid growth, with an average growth rate of  $\underline{0.56 \pm 0.013}$  Gt year-1, driven by industrialization, urbanization, and rising energy demand. In contrast, emissions become relatively stable from 2013 to 2016, with all inventories showing a slight decline ( $-0.07 \pm 0.022$  Gt year-1 on average). This short-term stagnation is mainly influenced by the adjustment of energy structure and industrial upgrades under China's 12th Five-Year Plan, and the implementation of air clean policy since 2013 (Han et al., 2020b; Shi et al., 2022; Zheng et al., 2025). ..."
- (3) **Section 4, paragraph 2:** "The six inventories in this study agree on three emission phases: a rapid increase of  $0.56 \pm 0.013$  Gt year-1 (2000–2013), a near-stagnation phase of  $-0.07 \pm 0.022$  Gt year-1 under the 12th Five-Year Plan and air clean policy (2013–2016), ..."

**References:**

Han, P., Zeng, N., Oda, T., Lin, X., Crippa, M., Guan, D., Janssens-Maenhout, G., Ma, X., Liu, Z., Shan, Y., Tao, S., Wang, H., Wang, R., Wu, L., Yun, X., Zhang, Q., Zhao, F., and Zheng, B.: Evaluating China's fossil-fuel CO2 emissions from a comprehensive dataset of nine inventories, Atmospheric Chemistry and Physics, 20, 11371–11385, https://doi.org/10.5194/acp-20-11371-2020, 2020.

Shi, Q., Zheng, B., Zheng, Y., Tong, D., Liu, Y., Ma, H., Hong, C., Geng, G., Guan, D., He, K., and Zhang, Q.: Co-benefits of CO2 emission reduction from China's clean air actions between 2013-2020, Nat Commun, 13, 5061, https://doi.org/10.1038/s41467-022-32656-8, 2022.

Zheng, L., Li, S., Hu, X., Zheng, F., Cai, K., Li, N., and Chen, Y.: Spatiotemporal comparative analysis of three carbon emission inventories in mainland China, Atmospheric Pollution Research, 16, 102417, https://doi.org/10.1016/j.apr.2025.102417, 2025.

**2. Figure 2:** Why are the results presented as a five-year average? The CEAD values for the period between 2012 and 2017 are occasionally negative. What factors could explain this phenomenon?

**Response:** We thank the reviewer for the question. The results in Figure 2 are presented as five-year averages to align with the global stocktake cycle defined by the Paris Agreement, which requires a comprehensive assessment every five years starting in 2023 (UNFCCC, 2015). Accordingly, we have adjusted the baseline year from 2002 to 2003, corresponding to the first global stocktake period. This revision is shown below.

Regarding the slightly negative CEADs growth rate during 2013–2018, this reflects a minor decline in CEADs emissions over that interval (from 10.14 Gt in 2013 to 9.77 Gt in 2018), consistent with the national emission stagnation driven by industrial restructuring under China's 12th Five-Year Plan and clean air policy. The five-year emission growth shown in Figure 2 is calculated as the difference between the final and initial year divided by five, which naturally yields a small negative value for CEADs in this period.

**Revision:**

Section 3.1, paragraph 4: "In response to the Paris Agreement's requirement of a global stocktake every five years starting in 2023 (https://unfccc.int/sites/default/files/paris\_agreement\_english\_.pdf), we analyze China's emissions variation every five years, using 2003 as the baseline year corresponding to the first global stocktake (Fig. 2). The highest growth is recorded in the period from 2003 to 2008 (> 0.52 Gt year 1) and 2008-2013 (> 0.45 Gt year 1), followed by a stable period in the years from 2013 to 2018, in which the CEADs even records a slight decline (-0.01 Gt year 1). Growth then resumed in 2018-2023, averaging 0.21 Gt year 1."

**Section 3.1, Figure 2:**

Figure 2. Average annual CO2 emission growth rate during the five-year periods.

**3. Figure 3:** has not been cited within the main text**

**Response:** Thank you for pointing out the citation issue for Figure 3. We have carefully checked our manuscript and confirmed that this figure is cited in *Section 3.1*, paragraph 5. The sentence reads: "... The sectoral CO2 emissions show that the electricity and heat production sector and the industry and construction sector dominate emissions and together account for over 78% of total emissions (Fig. 3). ...".

**4. Line 211:** Why was the year 2019 chosen as the reference year for comparing spatial patterns in Figure 4? Could this choice be clarified?**

**Response:** We thank the reviewer for the comment. As mentioned in Section 3.2.1, 2019 was chosen as the reference year because it is the most recent year for which all five gridded inventories (ODIAC, EDGAR, MEIC, CAMS, and GEMS) provide spatially explicit emission data. Moreover, 2019 represents a typical pre-pandemic year, unaffected by the COVID-19 lockdowns in 2020-2021 2019 is free from exceptional events such as the COVID-19 lockdowns, making it a representative baseline for comparison.

Although our manuscript focuses on 2019 due to space limitations, we also conducted preliminary analyses for the third emission phase (2016-2023). As illustrated in the GIF below, the spatial patterns of inter-inventory differences remain generally consistent over time, although the overall magnitude of emissions varying. The only notable exception occurs in the EDGAR–MEIC comparison, where differences in southwestern China shift from obvious positive to negative during 2016–2017. After 2017, the EDGAR–MEIC spatial differences stabilize, and other inventories relative to MEIC show minimal spatial variation throughout 2016–2023.

Temporal evolution of spatial differences in CO2 emissions between MEIC and other inventories (ODIAC, EDGAR, CAMS, and GEMS) during 2016–2023.

**5. Line 234:** Could the authors clarify why the MEIC inventory was chosen as the benchmark for Figure 5?

Response: We thank the reviewer for this valuable comment. Among the five gridded inventories (ODIAC, EDGAR, MEIC, CAMS, and GEMS) used in this study, both MEIC and GEMS are constructed using Chinese statistical data. Specifically, the energy consumption data in MEIC and GEMS are derived from the China Energy Statistical Yearbook (CESY) and the National Bureau of Statistics of China (NBS), respectively. Given that GEMS is a newly released dataset (2025) and MEIC has been developed and validated for more than a decade, we selected MEIC as the benchmark for comparison. MEIC is widely recognized and used when studying anthropogenic emissions in China. For example, it has been integrated into the MIX inventory as the Chinese component of the Asian anthropogenic emissions (Li et al., 2017) and was used to develop high-resolution (1 km × 1 km) emission maps for 2013 (Zheng et al., 2021). Previous studies have also shown that simulations based on MEIC are more consistent with observations than those using EDGAR or ODIAC in Beijing (Che et al., 2022) and perform better in Xianghe and Xinlong (Yang et al., 2025). We have revised our manuscript for clarifying the rationality of the benchmark choice.

**Revision:**

Section 3.2.1, paragraph 3: "To assess spatial consistency, we compared ODIAC, EDGAR, CAMS, and GEMS with MEIC as a benchmark (Fig. 5). MEIC was chosen because it is compiled using local statistics and has been widely applied and validated in previous studies (Li et al., 2017b; Zheng et al., 2021; Che et al., 2022; Yang et al., 2025), making it a reasonable reference for comparison. ..."

**References:**

Che, K., Cai, Z., Liu, Y., Wu, L., Yang, D., Chen, Y., Meng, X., Zhou, M., Wang, J., Yao, L., and Wang, P.: Lagrangian inversion of anthropogenic CO  $_2$  emissions from Beijing using differential column measurements, Environ. Res. Lett., 17, 075001, https://doi.org/10.1088/1748-9326/ac7477, 2022.

Li, M., Zhang, Q., Kurokawa, J., Woo, J.-H., He, K., Lu, Z., Ohara, T., Song, Y., Streets, D. G., Carmichael, G. R., Cheng, Y., Hong, C., Huo, H., Jiang, X., Kang, S., Liu, F., Su, H., and Zheng, B.: MIX: a mosaic Asian anthropogenic emission inventory under the international collaboration framework of the MICS-Asia and HTAP, Atmos. Chem. Phys., 17, 935–963, https://doi.org/10.5194/acp-17-935-2017, 2017.

Yang, H., Wu, K., Wang, T., Wang, P., and Zhou, M.: Atmospheric anthropogenic CO2 variations observed by tower in-situ measurements and simulated by the STILT model in the Beijing megacity region, Atmospheric Research, 325, 108258, https://doi.org/10.1016/j.atmosres.2025.108258, 2025.

Zheng, B., Cheng, J., Geng, G., Wang, X., Li, M., Shi, Q., Qi, J., Lei, Y., Zhang, Q., and He, K.: Mapping anthropogenic emissions in China at 1 km spatial resolution and its application in air quality modeling, Science Bulletin, 66, 612–620, https://doi.org/10.1016/j.scib.2020.12.008, 2021.

**6. Figure 7:** Could the authors clarify whether the pattern observed for Shanxi Province is unique to this province or if it occurs in other regions as well?

Response: We appreciate the reviewer's careful attention on the pattern observed in Shanxi Province. We generated a provincial heatmap showing the differences between CEADs (provinces) and CEADs (sectors). The provinces are sorted by provincial total emissions in descending order (Fig. S2). The results show that Shanxi is a clear outlier among all provinces, with differences exceeding 900 Mt CO2 year-1 after 2012, while differences in other provinces remain within 400 Mt year-1. While other provinces show spatially heterogeneous discrepancies, no other region exhibits a pattern of this magnitude. This unique characteristic is why we chose to highlight Shanxi Province in our analysis. Beyond Shanxi, Large differences (>100 Mt year-1) are mostly concentrated in provinces with higher total emissions (top 15 in 30 provinces), with few exceptions (e.g., Xinjiang in 2021). Provinces with lower total emissions (bottom 15) generally show smaller discrepancies (<50 Mt year-1), except for Xinjiang, Guizhou, and Ningxia. Overall, although the spatial pattern is heterogeneous, there is a general tendency for differences to decrease with provincial emission magnitude. We have added this provincial heatmap to the supplementary material and revised the manuscript accordingly.

**Revision:**

Section 3.3.1, paragraph 1: "CEADs provides two forms of CO2 emission estimates for provinces: the "province" series (referred to as CEADs (provinces)), which provides total emissions directly for each province, and the "sectors" series (referred to as CEADs (sectors)), which compiles fueland sector-specific emissions before summing them to the provincial totals. Significant discrepancies are observed between these two estimates in some provinces, with Shanxi emerging as a pronounced outlier. After 2012, the difference in Shanxi exceeds 900 Mt year-1, whereas in other provinces it remains below 400 Mt year-1 (Fig. S2). To investigate this divergence, we compare both CEADs estimates with other inventories in Shanxi (Fig. 7a). The results indicate that CEADs (provinces) exceeds CEADs (sectors) after 2008, ..."

**Section 7, Figure S2:**

Figure S2. Heatmap of the annual CO2 emission differences between CEADs (province) and CEADs (sector) for 30 Chinese provinces provided by CEADs during 2000–2021. Provinces are ordered by total emissions from highest to lowest.

---

## Author Comment (AC2)

**RC1**

This manuscript offers a robust comparative analysis of six CO2 emission inventories for China, integrating both local and global datasets. A key strength is its detailed assessment of spatial and temporal uncertainties, an often overlooked but policy-relevant aspect. The study contributes meaningfully by highlighting inventory discrepancies and emphasizing the importance of uncertainty assessments in emission reporting. However, I have the following specific comments that require clarification and revision before the manuscript can be considered for publication.

**General comments**

The manuscript is clearly written and well structured, with a logical flow that facilitates understanding of the main objectives and findings.

1. However, it is not entirely clear whether the emission inventories selected for analysis are the only relevant options available, or what criteria guided their selection. Since the manuscript references other inventories that were ultimately not included in the comparison, it would strengthen the study to provide a clearer rationale for the choices made.

Response: We sincerely thank the reviewer for this valuable comment regarding the selection criteria of the emission inventories. In this study, we aimed to ensure both temporal completeness and spatial representativeness when selecting inventories. The six inventories included (ODIAC2023, EDGAR2024, MEIC-global-CO2 v1.0, CAMS-GLOB-ANT v6.2, GEMS v1.0, and CEADs) provide continuous time-series covering most of the period from 2000 to 2023 (at least from 2000 to 2019 in GEMS) and have explicit coverage over mainland China. These inventories are also internationally recognized and widely cited in peer-reviewed studies (Li et al., 2017; Han et al., 2020; Liu et al., 2024; Zheng et al., 2025). Besides, they are freely available from official websites. Other inventories mentioned in the text, such as CHRED, were not included because their datasets are not directly accessible. Although the CHRED dataset has been partially integrated into the IPPU accounting platform (https://www.cityghg.com/toCauses?id=4), the platform only provides data for four discrete years (2005, 2010, 2015, and 2020), leaving substantial temporal gaps that prevent a consistent time-series analysis.

In this revision, we also added the national total CO2 emissions reported by the Chinese government in the National Greenhouse Gas Inventory (NGHGI) submitted to the UNFCCC (from documents China. 2024 Biennial Transparency Report (BTR). BTR1, and China. Biennial Update Report (BUR). BUR 4, available at <a href="https://unfccc.int/reports">https://unfccc.int/reports</a>). The NGHGI data are also temporally discontinuous, but provide 8 available years (2005, 2010, 2012, 2014, 2017, 2018, 2020, and 2021). The NGHGIs represent the officially reported values and therefore provide an independent benchmark to evaluate the consistency of the six bottom-up inventories. We have now clarified this rationale for the inventory selection in Section 2.

**Revision:**

- (1) Section 2, paragraph 1: "To ensure both temporal completeness and spatial representativeness, the selected emission inventories must provide a continuous time-series covering most of the 2000–2023 period (with at least 2000–2019 coverage in GEMS) and have explicit spatial coverage over mainland China. Six anthropogenic CO2 emission inventories, including five gridded inventories (ODIAC2023, EDGAR2024, MEIC-global-CO2 v1.0, CAMS v6.2, and GEMS v1.0) and one urban total emission inventory (CEADs), are applied to provide estimates of total emissions at the national, provincial, and city levels in China. As internationally recognized and widely used by previous studies (Li et al., 2017b; Han et al., 2020b; Liu et al., 2024; Zheng et al., 2025), these inventories are publicly available from official repositories."
- (2) Section 2, paragraph 2: "In addition to these six datasets, the National Greenhouse Gas Inventory (NGHGI) submitted by the Chinese government to the United Nations Framework Convention on Climate Change (UNFCCC, available at: https://unfccc.int/reports) was also collected. The NGHGI provides the officially reported national total emissions and therefore serves as an independent benchmark for evaluating the reliability of the six inventories. As NGHGI covers only discrete years (2005, 2010, 2012, 2014, 2017, 2018, 2020, and 2021), it is not included in the continuous temporal analysis but is used solely for national-level comparison."
- (3) **Section 2, paragraph 3:** "The specific information of the six selected inventories is presented in Section 2.1. ..."

**References:**

- Han, P., Zeng, N., Oda, T., Lin, X., Crippa, M., Guan, D., Janssens-Maenhout, G., Ma, X., Liu, Z., Shan, Y., Tao, S., Wang, H., Wang, R., Wu, L., Yun, X., Zhang, Q., Zhao, F., and Zheng, B.: Evaluating China's fossil-fuel CO2 emissions from a comprehensive dataset of nine inventories, Atmospheric Chemistry and Physics, 20, 11371–11385, https://doi.org/10.5194/acp-20-11371-2020, 2020.
- Li, M., Zhang, Q., Kurokawa, J., Woo, J.-H., He, K., Lu, Z., Ohara, T., Song, Y., Streets, D. G., Carmichael, G. R., Cheng, Y., Hong, C., Huo, H., Jiang, X., Kang, S., Liu, F., Su, H., and Zheng, B.: MIX: a mosaic Asian anthropogenic emission inventory under the international collaboration framework of the MICS-Asia and HTAP, Atmos. Chem. Phys., 17, 935–963, https://doi.org/10.5194/acp-17-935-2017, 2017.
- Liu, H., Hu, C., Xiao, Q., Zhang, J., Sun, F., Shi, X., Chen, X., Yang, Y., and Xiao, W.: Analysis of anthropogenic CO2 emission uncertainty and influencing factors at city scale in Yangtze River Delta region: One of the world's largest emission hotspots, Atmospheric Pollution Research, 15, 102281, https://doi.org/10.1016/j.apr.2024.102281, 2024.
- Zheng, L., Li, S., Hu, X., Zheng, F., Cai, K., Li, N., and Chen, Y.: Spatiotemporal comparative analysis of three carbon emission inventories in mainland China, Atmospheric Pollution Research, 16, 102417, https://doi.org/10.1016/j.apr.2025.102417, 2025.

2. The relevance of the topic is evident, especially in light of China's pivotal role in global emissions and its commitments under the Paris Agreement. Still, the manuscript would benefit from a more explicit explanation of why comparing the latest versions of these inventories is particularly important. A clearer articulation of what distinguishes this study from previous work (beyond simply the version updates) would improve accessibility, especially for readers less familiar with the topic.

Response: We thank the reviewer for this helpful comment. We have now revised the texts to more clearly state why using the latest inventory versions is essential and how this study differs from previous work. The latest versions incorporate updated activity data, emission factors, and spatial proxies, ensuring greater temporal completeness and accuracy. For example, ODIAC2023 incorporates the latest national fossil-fuel CO2 estimates from the CDIAC team (AppState, Gilfillan et al. 2021, Hefner and Marland, 2023), covering the period 2000–2022 (available at: <a href="https://db.cger.nies.go.jp/dataset/ODIAC/readme/readme\_2023\_20240605.txt">https://db.cger.nies.go.jp/dataset/ODIAC/readme/readme\_2023\_20240605.txt</a>). EDGAR2024 integrates updated activity data from IEA (2023) and FAO (2024), extends the time series of CO2 emissions to 2023 through a new "Fast Track" approach (Guizzardi et al., 2024; Crippa et al., 2024), and employs enhanced spatial proxies such as the Global Energy Monitor power plant dataset (available at: <a href="https://edgar.jrc.ec.europa.eu/dataset\_ghg2024">https://edgar.jrc.ec.europa.eu/dataset\_ghg2024</a>). These improvements significantly enhance temporal completeness and spatial accuracy compared to earlier versions (e.g., EDGAR v8.0, ODIAC2022).

We have also addressed our study's distinct contributions compared with earlier analyses (Han et al., 2020; L. Zheng et al., 2025). This work (1) extends the temporal coverage to 2000–2023 and identifies three distinct emission phases reflecting policy and energy structure changes; (2) evaluates inconsistencies within CEADs and recommends using CEADs (sectors) for provincial analyses; (3) reveals sectoral spatial allocation differences—especially between EDGAR and MEIC in the transport sector; (4) quantifies scale-dependent uncertainties, showing that provincial uncertainty (CV) is 2-10 times higher than national uncertainty; and (5) shows that CEADs and MEIC yield consistent estimates across nine representative provinces. At the national scale, CAMS shows the smallest deviation from the NGHGI, while ODIAC agrees most closely with the six-inventory mean during the study period. These revisions have been added to the Section 1 to highlight the rationale for using the latest inversions and to the Section 4 to summarize the new insights and methodological contributions.

**Revision**

- (1) **Section 1, paragraph 4:** "...Moreover, emission inventories are continuously updated to incorporate improved inputs (e.g., activity data, EFs, and refined methodology). Therefore, it is crucial to use the latest versions of the various inventories to capture these methodological updates and better understand the most recent patterns of China's anthropogenic CO2 emissions."
- (2) **Section 4, paragraph 5:** "In summary, this study extends previous work by identifying a three-phase trend in China's anthropogenic CO2 emissions from 2000 to 2023 and quantifying the emission uncertainties (1 $\sigma$ ) at both national and provincial levels. At the national level, CAMS shows the closest agreement with the government-reported NGHGI, while ODIAC aligns best with the multi-inventory mean over the study period. At the provincial level, the Chinese local inventories, CEADs and MEIC, provide the most consistent estimates for regional studies. Differences in spatial

proxies significantly affect the spatial distribution of sectoral emissions, as shown by the contrasting transport emission patterns in EDGAR and MEIC. We also clarify the appropriate use of CEADs for provincial analyses. Our results further underscore the importance of improving the consistency of regional inventories to provide a stronger scientific basis for China's emission mitigation and carbon neutrality policies."

3. The discussion of differences between inventories and their associated uncertainties is engaging and informative. However, a clear take-home message is lacking, particularly regarding which inventories may be considered more reliable or fit for specific purposes. While it is understandable that definitive recommendations may be difficult, the current conclusions are limited, with the mainly strong guidance being to avoid the provincial CEADs inventory. Offering more concrete insights or practical recommendations, especially in the context of supporting policymaking, would significantly strengthen the manuscript.

**Response:** We thank the reviewer for this constructive suggestion. We agree that identifying which inventories are more reliable is crucial. However, determining the accuracy of each inventory requires direct comparisons with independent observations (e.g., atmospheric CO2 measurements and inversion results), which is beyond the scope of this study. In this study, we focused on assessing the consistency among inventories and their deviations from independent references.

To strengthen the conclusions, we have now included the National Greenhouse Gas Inventory (NGHGI) data submitted by the Chinese government to the UNFCCC for comparison at the national level. We have revised Figure 1 to include NGHGI data. We assessed the consistency of the six inventories (2000–2023) by calculating mean absolute difference (MAD) of each inventory relative to the NGHGI and the six-inventory mean. Our findings show that CAMS exhibits the greatest consistency with the NGHGI, while ODIAC agrees most closely with the six-inventory mean.

At the provincial level, the uncertainties are 2-10 times higher than that at the national level. While these variations make it difficult to determine an absolute reference, our analysis (Section 3.2.2, paragraph 3) shows that CEADs and MEIC exhibit good agreement in nine representative provinces, particularly in Inner Mongolia, Shandong, Henan, Hubei, and Shanghai. We have revised Section 3.1 and Section 4 accordingly to clearly incorporate these quantitative consistency assessments and provide clearer practical insights

**Revision:**

(1) Section 3.1, paragraph 2: "To further assess the consistency of the six inventories, we calculate the mean absolute difference (MAD), which is defined as the multi-year mean of annual absolute differences between each inventory and either the NGHGI or the six-inventory mean. Compared with NGHGI, the MADs range from 0.156 Gt year-1 (CAMS) to 0.835 Gt year-1 (MEIC). Against the six-inventory mean, the MADs range from 0.12 Gt year-1 (ODIAC) to 0.449 Gt year-1 (MEIC). EDGAR reports the highest emissions, which is about 0.370 Gt year-1 larger than the mean emission. MEIC shows the lowest emission levels, which is about 0.449 Gt year-1 less than the mean emission. Overall, CAMS exhibits the greatest consistency with the NGHGI, being at least 30% lower than that of the other inventories. In comparison, ODIAC agrees most closely with the six-inventory mean, with an MAD at least 58% lower than the others."

- (2) **Section 4, paragraph 1:** "China's annual anthropogenic CO2 total emission increases from 3.42 Gt in 2000 to 12.03 Gt in 2023. When compared with the officially reported NGHGI and the six-inventory mean, CAMS shows the smallest deviation from the NGHGI, while ODIAC agrees most closely with the multi-inventory mean. The six inventories display a broadly consistent emission trend, but their discrepancies among the inventories have widened from 0.41 Gt year-1 to 1.63 Gt year-1, mainly due to the highest estimates reported from EDGAR and the lowest values estimated from MEIC, especially after 2012. ..."
- (3) **Section 4, paragraph 4:** "...The pronouncedly higher emissions in the coastal megacities (e.g., Shanghai, Jiangsu, and Guangdong) by ODIAC and the abnormal increase in CAMS by 50-230% in Liaoning, Hubei, and Shanghai exacerbate this divergence. Despite these inconsistencies, CEADs and MEIC exhibit broadly consistent estimates across nine provinces, especially in Inner Mongolia, Shandong, Henan, Hubei, and Shanghai."
- (4) Section 4, paragraph 5: "In summary, this study extends previous work by identifying a three-phase trend in China's anthropogenic CO₂ emissions from 2000 to 2023 and quantifying the emission uncertainties (1σ) at both national and provincial levels. At the national level, CAMS shows the closest agreement with the government-reported NGHGI, while ODIAC aligns best with the multi-inventory mean over the study period. At the provincial level, the Chinese local inventories, CEADs and MEIC, provide the most consistent estimates for regional studies. Differences in spatial proxies significantly affect the spatial distribution of sectoral emissions, as shown by the contrasting transport emission patterns in EDGAR and MEIC. We also clarify the appropriate use of CEADs for provincial analyses. Our results further underscore the importance of improving the consistency of regional inventories to provide a stronger scientific basis for China's emission mitigation and carbon neutrality policies."

**Section 3.1, Figure 1:**

Figure 1. Annual anthropogenic CO2 emissions in mainland China from 2000 to 2023, as reported by six emission inventories: EDGAR, MEIC, CAMS, CEADs (up to 2021), ODIAC (up to 2022), and GEMS (up to 2019), and one government-reported data (NGHGI). Apart from ODIAC, all inventories provide national totals directly. We calculated China's emissions by summing the grid values within China for ODIAC. The shaded area indicates the standard deviation of the six inventories. It's noteworthy that the inter-inventory mean and SD were calculated from the above mentioned six inventories.

**Specific Comments**

1. Line 35: To highlight China's role in global emissions, please include the percentage of China's anthropogenic emissions relative to global totals.

**Response:** We appreciate the reviewer's suggestion. According to the Global Carbon Project (GCP, 2024), China accounted for about 32% of global anthropogenic CO2 emissions in 2023. We have added this information in Introduction section to better emphasize China's role in global emissions.

**Revision:**

**Section 1, paragraph 1:** "...China, which is responsible for about 80% of East Asia's anthropogenic CO2 emissions (Xia et al., 2025) and about 32% of global CO2 emissions according to the Global Carbon Project (GCP, 2024; available at: https://globalcarbonbudget.org/), has committed to reaching peak emissions by 2030 and carbon neutrality by 2060. ..."

**2. Line 44: The CAMS inventory should be included in this overview for completeness.**

**Response:** We appreciate the reviewer's suggestion. The CAMS inventory has now been included in the revised manuscript.

**Revision:**

Section 1, paragraph 2: "...Global gridded products provide consistent, worldwide estimates with high spatial resolution (1 km or 0.1°), such as the Open-Data Inventory for Anthropogenic Carbon Dioxide (ODIAC) (Oda et al., 2018; Oda and Maksyutov, 2011), the Emissions Database for Global Atmospheric Research (EDGAR) (Janssens-Maenhout et al., 2019), the Global Emission Modeling System (GEMS) (Wang et al., 2013), and the Copernicus Atmosphere Monitoring Service (CAMS-GLOB-ANT, hereafter referred to as CAMS, Soulie et al., 2024). ..."

**3. Line 48: Are there specific reasons for not including CHRED in the analysis? Please clarify.**

**Response:** We thank the reviewer for this comment. Our primary selection criteria required inventories to provide a continuous time-series covering most of the 2000-2023 to ensure temporal completeness. The publicly accessible CHRED dataset (available at: <a href="https://www.cityghg.com/toCauses?id=4">https://www.cityghg.com/toCauses?id=4</a>) only provides data for four discrete years (2005, 2010, 2015, and 2020), which leaves substantial temporal gaps that prevent a consistent time-series analysis. We have added our selection criteria in the revised manuscript.

**Revision:**

Section 2, paragraph 1: "To ensure both temporal completeness and spatial representativeness, the selected emission inventories must provide a continuous time-series covering most of the 2000-2023 period (with at least 2000–2019 coverage in GEMS) and have explicit spatial coverage over mainland China. Six anthropogenic CO2 emission inventories, ..."

4. Line 80: Consider introducing the CAMS inventory definition earlier in this section alongside the others, for consistency.

**Response:** We appreciate this suggestion regarding the CAMS inventory definition. The CAMS definition has been introduced earlier in the revised manuscript to enhance consistency, as suggested in a previous review comment (Specific Comment 2). The CAMS-GLOB-ANT definition, including the abbreviation (CAMS), is now presented in Section 1, Paragraph 2.

5. Line 80: MEIC is initially described (line 47) as a China-specific inventory, but here it is treated as a global inventory. This inconsistency may confuse readers, particularly since line 116 clarifies that the global version of MEIC is used. Please harmonize these descriptions.

Response: We thank the reviewer for pointing out this potential confusion regarding the MEIC inventory. We acknowledge that the distinction between MEIC's China-specific and global products was not sufficiently clarified. The MEIC team produces two distinct CO2 emission products: a China-specific version (MEIC-China-CO2) and a global version (MEIC-Global-CO2). We selected the MEIC-Global-CO2 product v1.0 based on its two primary advantages: it offers a higher spatial resolution (0.1°×0.1°) compared to the then-latest MEIC-China-CO2 v1.4 (0.25°×0.25°), and its temporal coverage extends closer to the most recent years (1970–2023 vs 1970–2020). Importantly, while this product is globally scoped, the emissions calculation within the Chinese region retains the accuracy of a local inventory by using Chinese local energy statistics (from the China Energy Statistics Yearbook, CESY)) and emission factors (from the China Emission Accounts and Datasets, CEADs). We have revised content in Section 2.1, paragraph 3 to harmonize these descriptions and clarify that the global version was selected based on its superior technical specifications (spatial resolution and temporal coverage).

**Revision:**

**Section 2.1, paragraph 3:** "...MEIC uses the transportation network data from the China Digital Road Network Map (CDRM) to constrain the distribution of vehicle activity as well as population density, GDP, and land use for other sectors (Li et al., 2017a; Xu et al., 2024b). In this study, we use the latest MEIC-Global-CO2 product (v1.0), which provides higher spatial resolution (0.1° × 0.1°) and longer temporal coverage (1970-2023) than the MEIC-China-CO2 product (v1.4; 0.25° × 0.25°, up to 2020). It's noteworthy that although MEIC-Global-CO2 is a global product, its emissions calculations for China continue to rely on local energy statistics (CESY) and emission factors (CEADs), ensuring consistency with domestic data while improving spatiotemporal details."

6. Line 122: The mention of the number of species covered by CAMS is not relevant here, as the analysis focuses on a single species. Also, this level of detail is not provided for the other inventories.

**Response:** We thank the reviewer for this helpful comment. The description of the number of species covered by CAMS has been removed to maintain consistency with the level of detail provided for the other inventories.

**7. Line 198: Do you have any hypotheses as to why GEMS diverges from the trends observed in other inventories, especially in the residential and commercial sectors?**

**Response:** We thank the reviewer for this insightful question. We have further investigated the GEMS inventory and consulted with the dataset developers. The residential emissions provided by GEMS are considered more reliable, because the national residential emission survey for the Second National Pollution Source Census was conducted by the GEMS team. Even prior to the census, GEMS team had carried out a comprehensive, representative national survey. These surveys suggested that publicly available statistical sources (such as IEA and FAO) have underestimated the rapid transition of China's residential energy mix (Tao et al., 2018), which likely led to overestimated residential emissions in other inventories. We have revised the manuscript accordingly to clarify this point.

**Revision:**

Section 3.1, paragraph 5: "...while a reverse pattern was observed in GEMS. The residential emissions provided by GEMS are considered more reliable, as the national residential emission survey for the Second National Pollution Source Census was conducted by the GEMS team. Data from their surveys indicate that the publicly available statistical sources (such as the IEA and the Food and Agriculture Organization of the United Nations, FAO) have underestimated the rapid transition of China's residential energy mix (Tao et al., 2018), leading to likely overestimated residential emissions in other inventories. The changes in the size of sectoral CO2 emissions indicate the changes in China's energy structure and economic growth, highlighting the importance of incorporating locally based surveys for residential emissions to improve the accuracy of bottom-up inventories."

**Reference:**

Tao, S., Ru, M. Y., Du, W., Zhu, X., Zhong, Q. R., Li, B. G., Shen, G. F., Pan, X. L., Meng, W. J., Chen, Y. L., Shen, H. Z., Lin, N., Su, S., Zhuo, S. J., Huang, T. B., Xu, Y., Yun, X., Liu, J. F., Wang, X. L., Liu, W. X., Cheng, H. F., and Zhu, D. Q.: Quantifying the rural residential energy transition in China from 1992 to 2012 through a representative national survey, Nat Energy, 3, 567–573, https://doi.org/10.1038/s41560-018-0158-4, 2018.

8. Table 1: Time Resolution (GEMS column): Please change "Annually" to "Annual" to align with the other entries.

**Response:** The term "Annually" in the GEMS column of Table 1 has been corrected to "Annual" in the revised manuscript.

9. Table 1: Data Source row: Since the "last accessed" date is the same for all inventories, consider moving this note to a table footnote (e.g., marked with an asterisk) to streamline the table.

**Response:** The "last accessed" date has been moved to a table footnote to improve readability and streamline the presentation in Table 1.

10. Figure 3: The growth in electricity and heat production in CAMS appears to stabilize, unlike in other inventories where growth continues. Given CAMS is based on EDGAR, a similar trend would be expected. Could this discrepancy be linked to the use of CAMS-Tempo profiles?

Response: We thank the reviewer for this insightful comment. The stabilization of CO2 emissions in CAMS arises from its extrapolation approach. Specifically, CAMS uses EDGAR as the base dataset and applies growth factors (q) from the Community Emissions Data System (CEDS) to extend emissions beyond the final EDGAR year (Soulie et al. (2024)). Projected emissions follow exponential growth with base q. Because q values fluctuate around 1 (0.9-1.05), the extrapolated emissions exhibit minimal variation, resulting in nearly linear and stable trends. As shown in figure below (from CAMS official website), we think the CAMS-GLOB-ANT v6.2 used in this study builds on EDGAR v7 (up to 2021) and extrapolates emissions to 2026, showing similar post-2021 stabilization. This stabilization accounts for the flat trend in electricity and heat production in CAMS during 2021–2023. Moreover, the CAMS-GLOB-TEMPO profiles are only used to temporally disaggregate the annual CAMS-GLOB-ANT emissions into monthly values, not for extrapolation. We have clarified this in the revised Data and Methods section.

**Revision:**

**Section 2.1, paragraph 4:** "CAMS is a global inventory developed as part of the Copernicus Atmosphere Monitoring Service project. It builds on EDGAR and integrates several complementary datasets, including the Community Emissions Data System (CEDS) for the extrapolation of the emissions up to the current year, the CAMS-GLOB-TEMPO for monthly variability, and the CAMS-GLOB-SHIP for ship emissions. ..."

**1/ EDGARv7 Anthro CO2-excl-short-cycle yearly - 2/ CAMS-GLOB-ANT Anthro CO2-excl-short-cycle v6.2 yearly**

Time series of global anthropogenic CO2 emissions from EDGAR v7 and CAMS-GLOB-ANT v6.2 during 2000-2026 (source: https://eccad.sedoo.fr/#/data).

**Reference:**

Soulie, A., Granier, C., Darras, S., Zilbermann, N., Doumbia, T., Guevara, M., Jalkanen, J.-P., Keita, S., Liousse, C., Crippa, M., Guizzardi, D., Hoesly, R., and Smith, S. J.: Global anthropogenic emissions (CAMS-GLOB-ANT) for the Copernicus Atmosphere Monitoring Service simulations of air quality forecasts and reanalyses, Earth Syst. Sci. Data, 16, 2261–2279, https://doi.org/10.5194/essd-16-2261-2024, 2024.

**11. Line 212: It is unclear why MEIC is used as a benchmark for comparison. Please add a brief explanation of this choice.**

Response: We thank the reviewer for this valuable comment. Among the five gridded inventories (ODIAC, EDGAR, MEIC, CAMS, and GEMS) used in this study, both MEIC and GEMS are constructed using statistical data from the Chinese government and official departments. Specifically, the energy consumption data in MEIC and GEMS are derived from the China Energy Statistical Yearbook (CESY) and the National Bureau of Statistics of China (NBS), respectively. Given that GEMS is a newly released dataset (2025) and MEIC has been developed and validated for more than a decade, we selected MEIC as the benchmark for comparison. MEIC is widely recognized and used when studying anthropogenic emissions in China. For example, it has been integrated into the MIX inventory as the Chinese component of the Asian anthropogenic emissions (Li et al., 2017) and was used to develop high-resolution (1 km × 1 km) emission maps for 2013 (Zheng et al., 2021). Previous studies have also shown that simulations based on MEIC are more consistent with observations than those using EDGAR or ODIAC in Beijing (Che et al., 2022) and perform better in Xianghe and Xinlong (Yang et al., 2025). We have revised our manuscript for clarifying the rationality of the benchmark choice.

**Revision:**

Section 3.2.1, paragraph 3: "To assess spatial consistency, we compared ODIAC, EDGAR, CAMS, and GEMS with MEIC as a benchmark (Fig. 5). MEIC was chosen because it is compiled using local statistics and has been widely applied and validated in previous studies (Li et al., 2017b; Zheng et al., 2021; Che et al., 2022; Yang et al., 2025), making it a reasonable reference for comparison. ..."

**References:**

Che, K., Cai, Z., Liu, Y., Wu, L., Yang, D., Chen, Y., Meng, X., Zhou, M., Wang, J., Yao, L., and Wang, P.: Lagrangian inversion of anthropogenic CO 2 emissions from Beijing using differential column measurements, Environ. Res. Lett., 17, 075001, https://doi.org/10.1088/1748-9326/ac7477, 2022.

Li, M., Zhang, Q., Kurokawa, J., Woo, J.-H., He, K., Lu, Z., Ohara, T., Song, Y., Streets, D. G., Carmichael, G. R., Cheng, Y., Hong, C., Huo, H., Jiang, X., Kang, S., Liu, F., Su, H., and Zheng, B.: MIX: a mosaic Asian anthropogenic emission inventory under the international collaboration framework of the MICS-Asia and HTAP, Atmos. Chem. Phys., 17, 935–963, https://doi.org/10.5194/acp-17-935-2017, 2017.

Yang, H., Wu, K., Wang, T., Wang, P., and Zhou, M.: Atmospheric anthropogenic CO2 variations observed by tower in-situ measurements and simulated by the STILT model in the Beijing megacity region, Atmospheric Research, 325, 108258, https://doi.org/10.1016/j.atmosres.2025.108258, 2025.

Zheng, B., Cheng, J., Geng, G., Wang, X., Li, M., Shi, Q., Qi, J., Lei, Y., Zhang, Q., and He, K.: Mapping anthropogenic emissions in China at 1 km spatial resolution and its application in air quality modeling, Science Bulletin, 66, 612–620, https://doi.org/10.1016/j.scib.2020.12.008, 2021.

**12. Figure 5c: What accounts for the squared patches in this figure? A brief explanation in the caption or main text would help readers interpret the results.**

**Response:** We appreciate the reviewer's careful observation. The squared patches visible in Fig. 5c mainly occur in Xinjiang, Qinghai, Gansu, and Inner Mongolia. To verify their origin, we extracted CAMS emissions for these provinces, as shown in the figure below. The result shows that the squared patterns are inherent to the CAMS dataset itself. When analyzing spatial differences, only grid cells with valid values in both CAMS and MEIC were considered. Therefore, the spatial distribution of CAMS – MEIC in Fig. 5c directly reflects the pattern of CAMS emissions.

CAMS emission distribution in selected provinces (Xinjiang, Qinghai, Gansu, and Inner Mongolia).

13. Figures 4 & 5: In Figure 4, MEIC shows notable emissions over western China (green shading), while ODIAC does not. This difference should manifest as strong negative values (blue) in Figure 5, yet much of this area appears blank, which I assume represents NaN values. Did you apply any filtering? Please clarify.

**Response:** We thank the reviewer for this valuable comment. A spatial filter was applied before calculating the differences. Specifically, only grid cells with valid emission values in both inventories were retained for the difference maps. Grid cells containing NaN values in either dataset were excluded to ensure consistent comparison. As a result, areas where ODIAC has NaN values, such as parts of western China, appear blank in Fig. 5, even though MEIC reports valid emissions there.

14. Line 241: For clarity, please consider rephrasing this sentence, here is a suggestion: "Across the spatial domain, EDGAR generally reports lower emissions than MEIC, with negative differences prevailing throughout the region."

**Response:** The sentence has been revised as recommended to improve clarity.

**Revision:**

**Section 3.2.1, paragraph 4:** "Across the spatial domain, EDGAR generally reports lower emissions than MEIC, with negative differences prevailing throughout the region (Fig. 5b). ..."

**15. Line 287: Could the discrepancy in Shanxi be attributed to a specific sector? A sectoral analysis, as presented in the previous section, would be valuable here.**

**Response:** We thank the reviewer for this insightful comment. We examined CO2 emissions from CEADs (sectors) and CEADs (provinces) for Shanxi and found that the large discrepancy mainly arises from differences in raw coal–related emissions, which is the dominant contributor to total emissions (Wei, 2022). As shown in the figure below, CO2 emissions from raw coal in CEADs (provinces) are on average 664.71 Mt year-1 higher than those in CEADs (sectors), leading to an overall mean difference of 512.18 Mt year-1 between the two datasets. We have included this figure in the supplementary material and revised the manuscript to clarify the source of the discrepancy in Shanxi's CEADs emissions.

**Revision:**

Section 3.3.1, paragraph 1: "...In contrast, the CEADs (sectors) closely matches the other five independent inventories (ODIAC, EDGAR, MEIC, CAMS and GEMS), with its mean emissions deviating by no more than 3.84 Mt year-1 from the average of the five inventories. The large discrepancy between CEADs (provinces) and CEADs (sectors) mainly originates from the much higher raw coal-related emissions in CEADs (provinces) (Fig. S3), as coal is the dominant contributor to total emissions (Wei, 2022)."

**Section 7, Figure S3:**

Figure S3. Comparison of total CO2 emissions and raw coal—related CO2 emissions in Shanxi from CEADs (sectors) and CEADs (provinces) during 2000–2020. Solid lines represent total emissions, while dashed lines indicate emissions from raw coal combustion.

**Reference:**

Wei, C.: Historical trend and drivers of China's CO2 emissions from 2000 to 2020, Environ Dev Sustain, 1–20, https://doi.org/10.1007/s10668-022-02811-8, 2022.

16. Line 290: Could you comment on the provincial comparison of the two CEADs estimates beyond Shanxi? Do any provinces show consistent agreement between the two datasets, and are these primarily low-emission regions? A colored map showing the differences between the two CEADs estimates by province could be a useful addition

Response: We thank the reviewer for this constructive suggestion. We generated a provincial heatmap showing the differences between CEADs (provinces) and CEADs (sectors). The provinces are sorted by provincial total emissions in descending order (Fig. S2). The results show that Shanxi is a clear outlier, with differences exceeding 900 Mt CO2 year-1 after 2012, while differences in other provinces remain within 400 Mt year-1. Beyond Shanxi, the discrepancies are spatially heterogeneous and do not directly correspond to total provincial emissions. For instance, Guangdong (ranked fourth in total emissions) shows relatively small differences (<100 Mt year-1), whereas some mid-ranked provinces, such as Shaanxi (14th among 30 provinces), exhibit differences greater than 100 Mt year-1 in more than half of the years. Large differences (>100 Mt year-1) are mostly concentrated in provinces with higher total emissions, with few exceptions (e.g., Xinjiang in 2021). Provinces with lower total emissions generally show smaller discrepancies (<50 Mt year-1), except for Xinjiang, Guizhou, and Ningxia. Overall, although the spatial pattern is heterogeneous, there is a general tendency for differences to decrease with provincial emission magnitude. We have added this provincial heatmap to the supplementary material and revised the manuscript accordingly.

**Revision:**

Section 3.3.1, paragraph 1: "CEADs provides two forms of CO2 emission estimates for provinces: the "province" series (referred to as CEADs (provinces)), which provides total emissions directly for each province, and the "sectors" series (referred to as CEADs (sectors)), which compiles fueland sector-specific emissions before summing them to the provincial totals. Significant discrepancies are observed between these two estimates in some provinces, with Shanxi emerging as a pronounced outlier. After 2012, the difference in Shanxi exceeds 900 Mt year-1, whereas in other provinces it remains below 400 Mt year-1 (Fig. S2). To investigate this divergence, we compare both CEADs estimates with other inventories in Shanxi (Fig. 7a). The results indicate that CEADs (provinces) exceeds CEADs (sectors) after 2008, ..."

**Section 7, Figure S2:**

Figure S2. Heatmap of the annual CO2 emission differences between CEADs (province) and CEADs (sector) for 30 Chinese provinces provided by CEADs during 2000–2021. Provinces are ordered by total emissions from highest to lowest.

---

## Author Comment (AC3)

**RC3**

**General comments:**

This manuscript analyzes and compares six bottom-up inventories and assesses their uncertainty. This work compares different inventories from international and domestic teams, and it will be useful to the global stocktake and accurately assess China's CO2 emissions. The topic is interesting and meaningful, but many statements and explanations in the manuscripts are not rigorous enough. I suggest more modifications and improvements before acceptance.

**Special comments:**

**1. Is it reasonable to use the mean and SD to assess the uncertainty of these emission inventories?**

Response: We appreciate the reviewer's valuable question. Using the mean and standard deviation (SD) to assess inter-inventory variability is statistically reasonable and consistent with previous studies (Han et al., 2020; Li et al., 2017). Besides, the coefficient of variation (CV), calculated as SD/mean, is employed here to quantify uncertainties at both national and provincial scales. This metric has also been used in previous studies to assess the accuracy of emission-related activity data (Zhao et al., 2011) and determine CO2 mole fraction variations (Christian, 2018). Compared with SD alone, CV more effectively reflects the relative magnitude of variability with respect to the mean value.

**References:**

Christian, E.: Evaluation of Anthropogenic Carbon Dioxide (CO2) Concentrations along River Nworie, Imo State, Nigeria, Environment Pollution and Climate Change, https://doi.org/10.4172/2573-458X.1000159, 2018.

Han, P., Zeng, N., Oda, T., Lin, X., Crippa, M., Guan, D., Janssens-Maenhout, G., Ma, X., Liu, Z., Shan, Y., Tao, S., Wang, H., Wang, R., Wu, L., Yun, X., Zhang, Q., Zhao, F., and Zheng, B.: Evaluating China's fossil-fuel CO2 emissions from a comprehensive dataset of nine inventories, Atmospheric Chemistry and Physics, 20, 11371–11385, https://doi.org/10.5194/acp-20-11371-2020, 2020.

Li, M., Zhang, Q., Kurokawa, J., Woo, J.-H., He, K., Lu, Z., Ohara, T., Song, Y., Streets, D. G., Carmichael, G. R., Cheng, Y., Hong, C., Huo, H., Jiang, X., Kang, S., Liu, F., Su, H., and Zheng, B.: MIX: a mosaic Asian anthropogenic emission inventory under the international collaboration framework of the MICS-Asia and HTAP, Atmos. Chem. Phys., 17, 935–963, https://doi.org/10.5194/acp-17-935-2017, 2017.

Zhao, Y., Nielsen, C. P., Lei, Y., McElroy, M. B., and Hao, J.: Quantifying the uncertainties of a bottom-up emission inventory of anthropogenic atmospheric pollutants in China, Atmospheric

**2. Activity data and emission factors are the two important factors that influence the emission inventory. I also suggest adding this important information to Table 1, although point, line, and area source proxies are listed.**

**Response:** We thank the reviewer for this helpful suggestion. We agree that including information on activity data and emission factors is essential for understanding the basis of each inventory. Accordingly, we have added this information to Table 1, as shown below, to clearly indicate the data sources used by each inventory.

Table 1. Specification of emission inventory statistics.

|            | ODIAC     |    | EDGAR     |    | MEIC     |      | CAMS       | GEMS        | CEADs        |
|------------|-----------|----|-----------|----|----------|------|------------|-------------|--------------|
| Version    | ODIAC2023 |    | EDGAR2024 |    | v1.0     |      | v6.2       | v1.0        | NA           |
| Domain     | Global    |    | Global    |    | Global   |      | Global     | Global      | China        |
| Temporal   | 2000-2022 |    | 1970-2023 |    | 1970-202 | 23   | 2000-2026  | 1700-2019   | 1997-2021    |
| coverage   |           |    |           |    |          |      |            |             |              |
| Time       | Monthly   | or | Monthly   | or | Monthly  | or   | Monthly or | Monthly or  | Annual       |
| resolution | annual    |    | annual    |    | annual   |      | annual     | annual      |              |
| Activity   | CDIAC, BP |    | IEA       |    | CESY,    | IEA, | EDGAR,     | NBS, IEA    | CESY, NBS    |
| data       |           |    |           |    | BP       |      | CAMS-      |             |              |
|            |           |    |           |    |          |      | GLOB-      |             |              |
|            |           |    |           |    |          |      | Ship       |             |              |
| Emission   | IPCC      |    | IPCC      |    | CEADs,   |      | EDGAR      | Literature, | on-site      |
| factors    | tors      |    |           |    | national |      |            | on-site     | measurements |
|            |           |    |           |    | submissi | ons  |            | measureme   |              |
|            |           |    |           |    | in UNFO  | CCC, |            | nts         |              |
|            |           |    |           |    | IPCC     |      |            |             |              |

**3. The Chinese government also reports national greenhouse gas emissions to the UNFCCC. I think it is better to compare the national $CO_2$ emissions between government-reported data and the six bottom-up inventory data mentioned in this study.**

Response: We thank the reviewer for this helpful suggestion. We have now included the National Greenhouse Gas Inventory (NGHGI) data submitted by the Chinese government to the UNFCCC for national-level comparison. Figure 1 has been revised to incorporate the NGHGI data. To assess the consistency of the six inventories (2000–2023), we calculated the mean absolute difference (MAD) of each inventory relative to both the NGHGI and the six-inventory mean. The results show that CAMS exhibits the greatest consistency with the NGHGI, while ODIAC agrees most closely with the six-inventory mean. These additions help provide an independent benchmark for evaluating the overall agreement of the inventories at the national scale.

**Revision:**

- (1) Section 2, paragraph 2: "In addition to these six datasets, the National Greenhouse Gas Inventory (NGHGI) submitted by the Chinese government to the United Nations Framework Convention on Climate Change (UNFCCC, available at: https://unfccc.int/reports) was also collected. The NGHGI provides the officially reported national total emissions and therefore serves as an independent benchmark for evaluating the reliability of the six inventories. As NGHGI covers only discrete years (2005, 2010, 2012, 2014, 2017, 2018, 2020, and 2021), it is not included in the continuous temporal analysis but is used solely for national-level comparison."
- (2) Section 3.1, paragraph 2: "To further assess the consistency of the six inventories, we calculate the mean absolute difference (MAD), which is defined as the multi-year mean of annual absolute differences between each inventory and either the NGHGI or the six-inventory mean. Compared with NGHGI, the MADs range from 0.156 Gt year-1 (CAMS) to 0.835 Gt year-1 (MEIC). Against the six-inventory mean, the MADs range from 0.12 Gt year-1 (ODIAC) to 0.449 Gt year-1 (MEIC). EDGAR reports the highest emissions, which is about 0.370 Gt year-1 larger than the mean emission. MEIC shows the lowest emission levels, which is about 0.449 Gt year-1 less than the mean emission. Overall, CAMS exhibits the greatest consistency with the NGHGI, being at least 30% lower than that of the other inventories. In comparison, ODIAC agrees most closely with the six-inventory mean, with an MAD at least 58% lower than the others."
- (3) **Section 4, paragraph 1:** "China's annual anthropogenic CO2 total emission increases from 3.42 Gt in 2000 to 12.03 Gt in 2023. When compared with the officially reported NGHGI and the six-inventory mean, CAMS shows the smallest deviation from the NGHGI, while ODIAC agrees most closely with the multi-inventory mean. The six inventories display a broadly consistent emission trend, but their discrepancies among the inventories have widened from 0.41 Gt year-1 to 1.63 Gt year-1, ..."
- (4) **Section 4, paragraph 4:** "...The pronouncedly higher emissions in the coastal megacities (e.g., Shanghai, Jiangsu, and Guangdong) by ODIAC and the abnormal increase in CAMS by 50-230% in Liaoning, Hubei, and Shanghai exacerbate this divergence. Despite these inconsistencies, CEADs and MEIC exhibit broadly consistent estimates across nine provinces, especially in Inner Mongolia, Shandong, Henan, Hubei, and Shanghai."

**Section 3.1, Figure 1:**

Figure 1. Annual anthropogenic CO2 emissions in mainland China from 2000 to 2023, as reported by six emission inventories: EDGAR, MEIC, CAMS, CEADs (up to 2021), ODIAC (up to 2022), and GEMS (up to 2019), and one government-reported data (NGHGI). Apart from ODIAC, all inventories provide national totals directly. We calculated China's emissions by summing the grid values within China for ODIAC. The shaded area indicates the standard deviation of the six inventories. It's noteworthy that the inter-inventory mean and SD were calculated from the above mentioned six inventories.

4. Line 168-174. Many studies report that China's emissions peaked in 2013 or 2014, so the first phase is better set as 2000-2013 or 2014. Also, the second phase is mainly due to the air control policy, besides the adjustment of energy and industrial structure.

**Response:** We thank the reviewer for this valuable comment. It's important to identify the corresponding year of China's emissions peak. Accordingly, we have adjusted the phase division in our analysis to set the first phase as 2000–2013 and the second phase as 2013–2016. We also acknowledge that the emission stabilization during the second phase is influenced not only by energy structure adjustments and industrial upgrading under China's 12th Five-Year Plan, but also by the implementation of air clean policy since 2013 (Han et al., 2020b; Shi et al., 2022; Zheng et al., 2025). The corresponding text and linear regression statistics in the first and second emission phase has been revised for clarity.

**Revision:**

- (1) Abstract: "...The national total CO2 emissions increase from 3.43 (3.21–3.63) Gt year-1 in 2000 to 12.03 (11.35–12.98) Gt year-1 in 2023, with three growth periods: rapid growth (2000–2013,  $0.56\pm0.013$  Gt year-1), near-stagnation (2013–2016, -0.07±0.022 Gt year-1), ..."
- (2) **Section 3.1, paragraph 3:** "The increase in  $CO_2$  emissions shows three different phases (Fig. 1, Table 2). The first phase (2000–2013) shows the most rapid growth, with an average growth rate of  $0.56 \pm 0.013$  Gt year-1, driven by industrialization, urbanization, and rising energy demand. In

contrast, emissions become relatively stable from 2013 to 2016, with all inventories showing a slight decline ( $-0.07 \pm 0.022$  Gt year-1 on average). This short-term stagnation is mainly influenced by the adjustment of energy structure and industrial upgrades under China's 12th Five-Year Plan, and the implementation of air clean policy since 2013 (Han et al., 2020b; Shi et al., 2022; Zheng et al., 2025). ..."

- (3) **Section 4, paragraph 2:** "The six inventories in this study agree on three emission phases: a rapid increase of  $0.56 \pm 0.013$  Gt year-1 (2000–2013), a near-stagnation phase of  $-0.07 \pm 0.022$  Gt year-1 under the 12th Five-Year Plan and air clean policy (2013–2016), ..."
- 5. Line 180-185. Although the global stocktake is held every five years, the stocktake assesses the achievement of NDCs of each country. Also, the baseline year of the Chinese 2020 and 2030 carbon reduction targets is 2005. I suggest the authors rewrite these sentences.

**Response:** We thank the reviewer for this insightful suggestion. According to the *Paris Agreement* (Article 14; UNFCCC, 2015), the first global stocktake is scheduled for 2023, followed by subsequent assessments every five years. In light of this, we revised the text to clarify that our five-year interval analysis is designed to correspond with the global stocktake cycle rather than the Nationally Determined Contributions (NDC) baseline year of 2005. Accordingly, we use 2003 as the starting point for the first five-year assessment period and have rewritten the sentences to reflect this rationale.

**Revision:**

Section 3.1, paragraph 4: "In response to the Paris Agreement's requirement of a global stocktake every five years starting in 2023 (https://unfccc.int/sites/default/files/paris\_agreement\_english\_.pdf), we analyze China's emissions variation every five years, using 2003 as the baseline year corresponding to the first global stocktake (Fig. 2). The highest growth is recorded in the period from 2003 to 2008 (> 0.52 Gt year 1) and 2008-2013 (> 0.45 Gt year 1), followed by a stable period in the years from 2013 to 2018, in which the CEADs even records a slight decline (-0.01 Gt year 1). Growth then resumed in 2018-2023, averaging 0.21 Gt year 1."

**Section 3.1, paragraph 4:**

Figure 2. Average annual CO2 emission growth rate during the five-year periods.

**6. Figure 4. Point and line sources of CAMS originated from EDGAR (Table 1). Why is the line source information lost in Figure 4d, especially in western China? Furthermore, the map of means (Figure 4f), most of the line and area information was lost.**

Response: We thank the reviewer for this valuable comment. We have carefully examined the sectoral emissions of CAMS and found that the spatial gaps over western China are not due to missing line or point source data but rather to the absence of aviation emissions. Specifically, CAMS includes only three transportation subsectors—Off-road transportation, Road transportation, and Ships—but does not account for aircraft emissions. To verify this, we compared the spatial distributions of transportation emissions among EDGAR, CAMS, MEIC, and GEMS (ODIAC does not provide sectoral data). As shown in Figure S1 below, EDGAR, MEIC, and GEMS all display distinct emission patterns along major flight corridors over western China, while CAMS only shows road transport patterns. This confirms that the absence of aviation emissions in CAMS leads to the spatial gaps observed in that region. We have added this explanation to Section 3.2.1 to enhance clarity and integrity of our research.

Regarding the mean emission map (Fig. 4f), only grid cells with valid values in all inventories were included in the averaging. Therefore, regions appearing blank correspond mainly to areas where ODIAC lacks valid data, rather than to missing spatial information in the other inventories.

**Revision:**

Section 3.2.1, paragraph 1: "..., while regions with limited nighttime lighting, including both sparsely populated areas and areas with high population but limited lighting, such as Western Sichuan, Inner Mongolia, and Xinjiang, are not captured. By contrast, the spatial gaps over western China in CAMS (Fig. 4d) mainly arise from the lack of aviation emissions. CAMS accounts for transport emissions from road, off-road, and ships but omits aviation. As shown in Figure S1, EDGAR, MEIC, and GEMS capture distinct emission bands along major flight corridors over western China, whereas CAMS only shows the road transport pattern, explaining the missing emissions over western China."

**Section 7, Figure S1:**

Figure S1. Spatial distribution of CO2 emissions from transport sector in 2019 across four inventories (EDGAR, CAMS, MEIC, and GEMS).

**7. Figure 5c. Why are there some squares with high values in the west and northeast China?**

**Response:** We thank the reviewer for this insightful comment. As noted above, we only analyzed grid cells with valid values in both inventories. The squared patches visible in Fig. 5c mainly occur in Xinjiang, Qinghai, Gansu, and Inner Mongolia. To verify their origin, we extracted CAMS emissions for these provinces, as shown in the figure below. The results indicate that these squared patterns originate from the CAMS dataset itself.

CAMS emission distribution in selected provinces (Xinjiang, Qinghai, Gansu, and Inner Mongolia).

**8. Figure 7. Why is the CEADs province data nearly ten times higher than other inventories in Shanxi Province?**

**Response:** We thank the reviewer for this insightful comment. We examined CO2 emissions from CEADs (sectors) and CEADs (provinces) for Shanxi and found that the large discrepancy mainly arises from differences in raw coal–related emissions, which is the dominant contributor to total emissions (Wei, 2022). As shown in the figure below, CO2 emissions from raw coal in CEADs (provinces) are on average 664.71 Mt year-1 higher than those in CEADs (sectors), leading to an overall mean difference of 512.18 Mt year-1 between the two datasets. We have included this figure in the supplementary material and revised the manuscript to clarify the source of the discrepancy in Shanxi's CEADs emissions.

**Revision:**

(1) **Section 3.3.1, paragraph 1:** "...In contrast, the CEADs (sectors) closely matches the other five independent inventories (ODIAC, EDGAR, MEIC, CAMS and GEMS), with its mean emissions deviating by no more than 3.84 Mt year-1 from the average of the five inventories. The large discrepancy between CEADs (provinces) and CEADs (sectors) mainly originates from the much higher raw coal–related emissions in CEADs (provinces) (Fig. S3), as coal is the dominant contributor to total emissions (Wei, 2022)."

**Section 7, Figure S3:**

Figure S3. Comparison of total CO2 emissions and raw coal–related CO2 emissions in Shanxi from CEADs (sectors) and CEADs (provinces) during 2000–2020. Solid lines represent total emissions, while dashed lines indicate emissions from raw coal combustion.

**Reference:**

Wei, C.: Historical trend and drivers of China's CO2 emissions from 2000 to 2020, Environ Dev Sustain, 1–20, https://doi.org/10.1007/s10668-022-02811-8, 2022.

9. Figure 8. EAGAR and MEIC are the highest and lowest inventories for national CO2 emissions, but these values varied at the provincial level. What are the key factors that affected these results? For example, CAMS had the highest values in Liaoning, Hubei provinces and Shanghai but the lowest in Hebei and Shandong provinces.

**Response:** We appreciate the reviewer's insightful question regarding the provincial variations among inventories. The inconsistency between EDGAR and MEIC at the national and provincial scales likely arises from their different downscaling methods. Differences in spatial proxies can significantly affect the spatial distribution of sectoral emissions, as illustrated by the contrasting transport emission patterns in EDGAR and MEIC (Section 3.2.2).

Although CAMS uses EDGAR emission data as its primary foundation, it also incorporates additional spatial proxies like CAMS-GLOB-Ship for sectoral allocation (Soulie et al., 2024). Consequently, CAMS may assign relatively higher emissions to provinces like Liaoning, Hubei, and Shanghai, while allocating lower values to Hebei and Shandong, depending on how industrial, transport, and energy-use proxies are spatially represented. Therefore, while inventories may show consistent national totals, differences in spatial proxy selection and downscaling methods can lead

to noticeable discrepancies at the provincial scale.

**Reference:**

Soulie, A., Granier, C., Darras, S., Zilbermann, N., Doumbia, T., Guevara, M., Jalkanen, J.-P., Keita, S., Liousse, C., Crippa, M., Guizzardi, D., Hoesly, R., and Smith, S. J.: Global anthropogenic emissions (CAMS-GLOB-ANT) for the Copernicus Atmosphere Monitoring Service simulations of air quality forecasts and reanalyses, Earth Syst. Sci. Data, 16, 2261–2279, https://doi.org/10.5194/essd-16-2261-2024, 2024.

**10. Figures S1&S2, Why do SD and CV for Hubei and Guangdong decrease sharply in 2023?**

**Response:** Thank you for your attention on this detail. We carefully rechecked the original provincial emission data and confirmed that there are no calculation or processing errors. The sharp decrease in the CV of Hubei and Guangdong in 2023 mainly results from the reduced number of available inventories. Specifically, by 2023, only three inventories—EDGAR, MEIC, and CAMS—provided data, whereas ODIAC, CEADs, and GEMS had ended earlier (GEMS in 2019, CEADs in 2021, and ODIAC in 2022).

As shown in Figures 8c and 8e, ODIAC consistently reported the highest emissions in Guangdong and the lowest in Hubei. The absence of ODIAC in 2023 therefore reduces the spread among inventories, leading to markedly lower SD and CV values in these two provinces. To illustrate this data-coverage effect, we have added shading in Figure S5 to indicate the years after 2019, when the number of available inventories began to decline.

**Revision: Section 7, Figure S5:**

Figure S5. Coefficient of variation (CV) of emissions at national level and for nine typical provinces during 2000-2023. The shaded area represents the period after 2019, when the number of available emission inventories began to decrease (GEMS ended in 2019, CEADs in 2021, and ODIAC in 2022).

**11. Table 1. Why can CAMS report the data in 2026 when it was published in 2023?**

**Response:** We appreciate the reviewer's insightful comment. According to Soulie et al. (2024), CAMS uses EDGAR data as its primary input and applies the Community Emissions Data System (CEDS) to extrapolate emissions to recent years. In our analysis, CAMS v6.2 is based on EDGAR v7 (up to 2021) and extends the emissions estimates up to 2026. We have updated the Data and Methods section to clarify this point.

**Revision:**

**Section 2.1, paragraph 4:** "CAMS is a global inventory developed as part of the Copernicus Atmosphere Monitoring Service project. It builds on EDGAR and integrates several complementary datasets, including the Community Emissions Data System (CEDS) for the extrapolation of the emissions up to the current year, the CAMS-GLOB-TEMPO for monthly variability, and the CAMS-GLOB-SHIP for ship emissions. ..."

**Reference:**

Soulie, A., Granier, C., Darras, S., Zilbermann, N., Doumbia, T., Guevara, M., Jalkanen, J.-P., Keita, S., Liousse, C., Crippa, M., Guizzardi, D., Hoesly, R., and Smith, S. J.: Global anthropogenic emissions (CAMS-GLOB-ANT) for the Copernicus Atmosphere Monitoring Service simulations of air quality forecasts and reanalyses, Earth Syst. Sci. Data, 16, 2261–2279, https://doi.org/10.5194/essd-16-2261-2024, 2024.

**12. Line 104. What does "BP plc" mean?**

**Response:** We thank the reviewer for the question. "BP plc" refers to BP p.l.c., formerly known as British Petroleum. The company later adopted the abbreviation "BP" as its official name. It is a global energy company that publishes the BP Statistical Review of World Energy, which provides widely used energy activity data. This information has been added to the revised manuscript.

**Revision:**

**Section 2.1, paragraph 1:** "...such as BP plc (formerly the British Petroleum company p.l.c.), the United States Geological Survey (USGS), ..."

---

## Author Comment (AC4)

**RC4**

The manuscript by Yang et al. (2025) addresses an important issue in carbon emission reporting, namely the comparison of six different bottom-up inventories using China as a case study. Accurate quantification of CO2 emissions is critical for developing effective mitigation policies. The authors' approach of including three global inventories and three local inventories makes the comparison meaningful and comprehensive. The manuscript is clearly written and was enjoyable to read. I have the following specific comments that require clarification before the manuscript can be considered for publication

**Specific Comments:**

**Introduction section**

I found the introduction engaging, but a few aspects could be elaborated further:

1. Please clarify why China was selected as the case study. Is it solely because China is the world's second largest emitter of CO2, or also because it provides a unique combination of global and local inventories suitable for comparison? Additionally, given that many similar studies have already been conducted for China, does this choice facilitate comparison with existing literature? Please specify.

Response: We thank the reviewer for this thoughtful question regarding the selection of China as our case study. China was chosen for both scientific and practical reasons. Scientifically, China accounts for approximately 80% of East Asia's anthropogenic CO2 emissions (Xia et al., 2025) and about 32% of global emissions according to the Global Carbon Project (GCP, 2024; available at <a href="https://globalcarbonbudget.org/">https://globalcarbonbudget.org/</a>). Practically, China has pledged to peak its CO2 emissions by 2030 or earlier and to reduce CO2 intensity by 60–65% relative to 2005 levels (SCIO, The State Council Information Office of China; available at <a href="http://www.scio.gov.cn/">http://www.scio.gov.cn/</a>). Accurate quantification of China's emissions is therefore critical for understanding its carbon budget and for supporting national mitigation policies.

Second, China was selected because its energy structure is undergoing an obvious transition to achieve the dual-carbon targets. This transformation is being driven by policies such as the renewable portfolio standards (RPS) and the clean air policy, which have promoted the adjustment of energy structure and industrial upgrades. The share of renewable energy in China's total power generation increased from 16.6% in 2000 to 28.2% in 2020, reflecting steady progress toward cleaner energy sources. However, fossil fuels still dominate the mix, and issues such as overcapacity in energy supply remain (Zhao et al., 2022). Therefore, assessing anthropogenic CO2 emissions under this transitional energy structure is crucial for evaluating the effectiveness of China's mitigation efforts.

Furthermore, although many previous studies have analyzed China's CO2 emissions, our work extends the temporal coverage (2000–2023) beyond earlier analyses (e.g., Han et al., 2000–2016;

Zheng et al., 2006–2021) by incorporating the latest versions of six major inventories. This design enables both temporal and methodological comparison with prior research, refining the understanding of inter-inventory discrepancies and uncertainties. For example, our analysis identifies three distinct emission phases, quantifies national and provincial uncertainties ( $1\sigma$ ), and shows that EDGAR estimates the highest national emissions and MEIC the lowest, differing from the near-agreement reported by Han et al. (2020b). Collectively, these advances allow a more robust evaluation of how inventory methodologies and consistency have evolved over time.

We have revised the Introduction to emphasize the significance and rationale for studying China's emissions.

**Revision:**

Section 1, paragraph 1: "China, which is responsible for about 80% of East Asia's anthropogenic CO2 emissions (Xia et al., 2025) and about 32% of global CO2 emissions according to the Global Carbon Project (GCP, 2024; available at: https://globalcarbonbudget.org/), has committed to reaching peak emissions by 2030 and carbon neutrality by 2060. Besides, China's energy structure is also undergoing an obvious transition driven by policies such as the renewable portfolio standards (RPS) and the clean air policy, which promote cleaner energy and industrial upgrades. The share of renewables in total power generation increased from 16.6% in 2000 to 28.2% in 2020, although fossil fuels still dominate and overcapacity issues remain (Zhao et al., 2022). Under this ongoing energy transition, accurate quantification of anthropogenic CO2 emissions and understanding the uncertainties in emissions inventories are needed to guide emission reduction policies toward the dual-carbon goals (Li et al., 2017a)."

**References:**

Li, M., Liu, H., Geng, G., Hong, C., Liu, F., Song, Y., Tong, D., Zheng, B., Cui, H., Man, H., Zhang, Q., and He, K.: Anthropogenic emission inventories in China: a review, National Science Review, 4, 834–866, https://doi.org/10.1093/nsr/nwx150, 2017.

Xia, L., Liu, R., Fan, W., and Ren, C.: Emerging carbon dioxide hotspots in East Asia identified by a top-down inventory, Commun Earth Environ, 6, 1–13, https://doi.org/10.1038/s43247-024-01991-7, 2025.

Zhao, F., Bai, F., Liu, X., and Liu, Z.: A Review on Renewable Energy Transition under China's Carbon Neutrality Target, Sustainability, 14, 15006, https://doi.org/10.3390/su142215006, 2022.

2. The authors have summarized previous studies from China that compared a few inventories. What is the novelty of the present work? Is the use of updated versions of inventories the only advancement, or are there other new aspects? Please state this explicitly.

**Response:** We thank the reviewer for this important question regarding the novelty of our study. Beyond the use of updated inventory versions, our work introduces several key advancements. Specifically, it (1) extends the temporal coverage to 2000–2023, identifying three distinct emission phases linked to China's evolving energy policy and industrial structure; (2) evaluates the internal consistency of CEADs data and recommends prioritizing CEADs (sectors) for provincial analyses;

(3) reveals notable sectoral spatial allocation discrepancies, particularly between EDGAR and MEIC in the transport sector; and (4) quantifies scale-dependent uncertainties, showing that provincial uncertainties are two to ten times higher than at national level. We have added a short paragraph in the Introduction and a more detailed paragraph in the Conclusion explicitly outlining the main advancements compared with previous studies.

**Revision:**

- (1) Section 1, paragraph 5: "To this aim, this study conducts a comprehensive analysis of the spatiotemporal variation of China's anthropogenic CO2 emissions and investigates the differences among six widely used emission inventories at their latest versions: the global inventories ODIAC, EDGAR, MEIC, GEMS, CAMS, and the China-specific inventory CEADs. The data and methods are presented in Section 2. We report our results in Section 3 and conclude the paper in Section 4. Compared with previous studies (Han et al., 2020b; Zheng et al., 2025), we extend the temporal coverage to 2000-2023, enabling a more current and consistent assessment of recent emission trends, inter-inventory discrepancies, and scale-dependent uncertainties across China."
- (2) Section 4, paragraph 5: "In summary, this study extends previous work by identifying a three-phase trend in China's anthropogenic CO2 emissions from 2000 to 2023 and quantifying the emission uncertainties (1σ) at both national and provincial levels. At the national level, CAMS shows the closest agreement with the government-reported NGHGI, while ODIAC aligns best with the multi-inventory mean over the study period. At the provincial level, the Chinese local inventories, CEADs and MEIC, provide the most consistent estimates for regional studies. Differences in spatial proxies significantly affect the spatial distribution of sectoral emissions, as shown by the contrasting transport emission patterns in EDGAR and MEIC. We also clarify the appropriate use of CEADs for provincial analyses. Our results further underscore the importance of improving the consistency of regional inventories to provide a stronger scientific basis for China's emission mitigation and carbon neutrality policies."

**Result Section**

**Section 3.1**

The authors state that differences among the emission inventories become more pronounced after 2012 and continue to diverge in recent years. However, the manuscript does not provide an explanation for this trend. It would greatly benefit the reader if the authors elaborated on the possible reasons for this divergence—for example, changes in activity data sources, revisions in statistical reporting, or methodological updates within specific inventories. Such context is essential to help readers better understand the evolution of Chinese emissions estimates over time.

**Response:** We thank the reviewer for this constructive comment. As noted in Section 3.1, the post-2012 divergence among inventories is mainly driven by EDGAR reporting the highest emissions and MEIC the lowest. We further investigated the possible reasons for this behavior by comparing the versions used in our study (EDGAR 2024 and MEIC-Global-CO2 v1.0) with those used by Han et al. (2020b) (EDGAR v4.3.2 and MEIC v1.3). Our analysis shows that EDGAR's national totals remain almost unchanged between the two versions, whereas MEIC-Global-CO2 v1.0 reports

significantly lower emissions than MEIC v1.3 (by about 1.43 Gt year-1 on average over 2008–2017). Consequently, the increased inter-inventory divergence after 2013 primarily originates from the downward revision in the latest MEIC dataset. Since the MEIC team does not provide detailed documentation on version-specific updates publicly, we can only infer that this reduction may reflect changes in energy statistics, emission factors, and data processing procedures introduced in the latest MEIC product. We have clarified this explanation in the revised manuscript to help readers better interpret the divergence among inventories after 2012.

**Revision:**

Section 4, paragraph 1: "China's annual anthropogenic CO2 total emission increases from 3.42 Gt in 2000 to 12.03 Gt in 2023. When compared with the officially reported NGHGI and the six-inventory mean, CAMS shows the smallest deviation from the NGHGI, while ODIAC agrees most closely with the multi-inventory mean. The six inventories display a broadly consistent emission trend, but their discrepancies among the inventories have widened from 0.41 Gt year-1 to 1.63 Gt year-1, mainly due to the highest estimates reported from EDGAR and the lowest values estimated from MEIC, especially after 2012. Our results are consistent with Zheng et al. (2025) but opposite to Han et al. (2020b), demonstrating the differences in emission versions (Our study: EDGAR2024, MEIC-global-CO2 v1.0; Zheng: EDGAR v7.0, MEIC-China-CO2 v1.4; Han: EDGAR v4.3.2, MEIC-China-CO2 v1.3). A comparison between these versions (Fig. S6) shows that the divergence mainly arises from a downward revision in the latest MEIC dataset, which reports about 1.43 Gt year-1 lower emissions on average over 2008–2017. In contrast, EDGAR's national totals remained nearly unchanged across versions, with differences within 0.001 Gt year-1 during 2000-2012. These results highlight the significant impact of inventory version updates on comparative emission analyses."

**Section 7, Figure S6:**

Figure S6. Comparison of national CO2 emissions from different versions of the EDGAR and MEIC inventories. The older versions (EDGAR v4.3.2 and MEIC-China-CO2 v1.3) used in Han et al. (2020b) are compared with the updated versions (EDGAR 2024 and MEIC-Global-CO2 v1.0) used in this study.

**Conclusion section**

The conclusion could be strengthened by addressing the following points:

**1. What is the main take-home message from this study?**

Response: Thanks for the comment. The key findings can be summarized as follows: (1) China's anthropogenic CO2 emissions from 2000–2023 exhibit three distinct growth phases driven by changes in energy policy and structure; (2) CEADs (sectors) provides more consistent estimates than CEADs (provinces) at both provincial level and national level; (3) large spatial discrepancies among inventories originate mainly from different downscaling proxies and spatial allocation approaches, as illustrated by the contrasting spatial pattern between EDGAR and MEIC, and the inter-inventory discrepancies at the provincial level; (4) provincial level uncertainties are substantially higher (2-10 times) than national ones (5) CEADs and MEIC yield consistent estimates across nine representative provinces. At the national scale, CAMS shows the smallest deviation from the National Greenhouse Gas Inventory (NGHGI), while ODIAC aligns most closely with the six-inventory mean during the study period. These clarifications have been added to Section 4 to summarize the new insights contributions.

**Revision:**

Section 4, paragraph 5: "In summary, this study extends previous work by identifying a three-phase trend in China's anthropogenic CO2 emissions from 2000 to 2023 and quantifying the emission uncertainties (1 a) at both national and provincial levels. At the national level, CAMS shows the closest agreement with the government-reported NGHGI, while ODIAC aligns best with the multi-inventory mean over the study period. At the provincial level, the Chinese local inventories, CEADs and MEIC, provide the most consistent estimates for regional studies. Differences in spatial proxies significantly affect the spatial distribution of sectoral emissions, as shown by the contrasting transport emission patterns in EDGAR and MEIC. We also clarify the appropriate use of CEADs for provincial analyses. Our results further underscore the importance of improving the consistency of regional inventories to provide a stronger scientific basis for China's emission mitigation and carbon neutrality policies."

**2. Which inventory performs better overall for China?**

**3. Are certain inventories more reliable in high-emission regions versus low-emission regions?**

Response to Comments 2 and 3: We thank the reviewer for these constructive questions regarding the relative reliability and regional performance of different inventories. Determining which inventory performs best requires evaluation against independent observation-based datasets (e.g., atmospheric CO2 measurements and inversion results), which is beyond the scope of this study. Instead, our analysis focuses on assessing internal consistency among inventories and their deviations from available references.

To strengthen the conclusions, we have now included the National Greenhouse Gas Inventory

(NGHGI) submitted by the Chinese government to the UNFCCC as a national benchmark. Figure 1 has been updated accordingly. Consistency was assessed by calculating the mean absolute difference (MAD) of each inventory relative to both the NGHGI and the six-inventory mean. The results indicate that CAMS shows the best agreement with the NGHGI, while ODIAC aligns most closely with the six-inventory mean throughout 2000–2023.

At the provincial level, uncertainties are two to ten times larger than those at the national scale, making it difficult to identify a single "best" inventory. Nonetheless, our analysis (Section 3.2.2) shows that CEADs and MEIC exhibit strong agreement across nine representative provinces, particularly in Inner Mongolia, Shandong, Henan, Hubei, and Shanghai. These findings have been incorporated into Sections 3.1 and 4 to provide clearer, quantitative insights into inventory reliability across different spatial scales and emission intensities.

**Revision:**

- (1) Section 3.1, paragraph 2: "To further assess the consistency of the six inventories, we calculate the mean absolute difference (MAD), which is defined as the multi-year mean of annual absolute differences between each inventory and either the NGHGI or the six-inventory mean. Compared with NGHGI, the MADs range from 0.156 Gt year-1 (CAMS) to 0.835 Gt year-1 (MEIC). Against the six-inventory mean, the MADs range from 0.12 Gt year-1 (ODIAC) to 0.449 Gt year-1 (MEIC). EDGAR reports the highest emissions, which is about 0.370 Gt year-1 larger than the mean emission. MEIC shows the lowest emission levels, which is about 0.449 Gt year-1 less than the mean emission. Overall, CAMS exhibits the greatest consistency with the NGHGI, being at least 30% lower than that of the other inventories. In comparison, ODIAC agrees most closely with the six-inventory mean, with an MAD at least 58% lower than the others."
- (2) **Section 4, paragraph 1:** "China's annual anthropogenic CO2 total emission increases from 3.42 Gt in 2000 to 12.03 Gt in 2023. When compared with the officially reported NGHGI and the six-inventory mean, CAMS shows the smallest deviation from the NGHGI, while ODIAC agrees most closely with the multi-inventory mean. The six inventories display a broadly consistent emission trend, but their discrepancies among the inventories have widened from 0.41 Gt year-1 to 1.63 Gt year-1, ..."
- (3) **Section 4, paragraph 4:** "...The pronouncedly higher emissions in the coastal megacities (e.g., Shanghai, Jiangsu, and Guangdong) by ODIAC and the abnormal increase in CAMS by 50-230% in Liaoning, Hubei, and Shanghai exacerbate this divergence. Despite these inconsistencies, CEADs and MEIC exhibit broadly consistent estimates across nine provinces, especially in Inner Mongolia, Shandong, Henan, Hubei, and Shanghai."
- (4) Section 4, paragraph 5: "In summary, this study extends previous work by identifying a three-phase trend in China's anthropogenic CO2 emissions from 2000 to 2023 and quantifying the emission uncertainties (1σ) at both national and provincial levels. At the national level, CAMS shows the closest agreement with the government-reported NGHGI, while ODIAC aligns best with the multi-inventory mean over the study period. At the provincial level, the Chinese local inventories, CEADs and MEIC, provide the most consistent estimates for regional studies. Differences in spatial proxies significantly affect the spatial distribution of sectoral emissions, as shown by the contrasting transport emission patterns in EDGAR and MEIC. We also clarify the appropriate use of CEADs for provincial analyses. Our results further underscore the importance of improving the consistency of regional inventories to provide a stronger scientific basis for China's emission mitigation and

**Section 3.1, Figure 1:**

Figure 1. Annual anthropogenic CO2 emissions in mainland China from 2000 to 2023, as reported by six emission inventories: EDGAR, MEIC, CAMS, CEADs (up to 2021), ODIAC (up to 2022), and GEMS (up to 2019), and one government-reported data (NGHGI). Apart from ODIAC, all inventories provide national totals directly. We calculated China's emissions by summing the grid values within China for ODIAC. The shaded area indicates the standard deviation of the six inventories. It's noteworthy that the inter-inventory mean and SD were calculated from the above mentioned six inventories.

Currently, these questions remain unanswered. I think including these aspects will be helpful for readers, providing them with clearer guidance and enhancing the practical value of the study.

**Recommendation:** This manuscript has the merit and it presents valuable data. However, it requires above minor revisions to be addressed before considered for the publication in Atmospheric Chemistry and Physics journal.

---

## Author Comment (AC5)

**RC2**

The paper compares six CO2 emission inventories for China from 2000 to 2023, including global inventories (ODIAC, EDGAR, GEMS) and China-specific ones (MEIC, CHRED, CEADs). It highlights large differences between inventories, especially EDGAR vs. MEIC, and differences in spatial distributions. This is important because China has ambitious carbon reduction goals, so accurate quantification of CO2 emissions is essential for policy and climate modelling. The paper fits within the journal's scope as it addresses atmospheric emissions and their uncertainties.

Limitations of this review: I am not an expert in CO2 emissions inventories and the relevant literature, so my comments focus on interpretation, clarity, and presentation rather than technical accuracy of methods.

**Major comments**

The paper is well-structured, the argument is easy to follow, and the language is clear. However, the following aspects would need to be addressed before publication.

**1. Clarify the novelty of the study**

It is unclear how this work differs from previous studies. Is the novelty in using updated versions of inventories, applying new harmonisation methods, or drawing new conclusions? Please add a short paragraph in the introduction explicitly stating what is new compared to other studies mentioned (e.g., Han et al., 2020a; Liu et al., 2015; L. Zheng et al., 2025).

Response: We appreciate the reviewer's valuable comment. To highlight the novelty of our work, we have added a paragraph in the Introduction and a more detailed paragraph in the Conclusion explicitly outlining the main advancements compared with previous studies. Specifically, this study (1) extends the temporal coverage to 2000–2023 and identifies three distinct emission phases reflecting changes in energy policy and structure; (2) evaluates internal inconsistencies within CEADs and recommends using CEADs (sectors) for provincial analyses; (3) reveals significant sectoral spatial allocation differences, particularly between EDGAR and MEIC in the transport sector; (4) quantifies scale-dependent uncertainties, showing that provincial uncertainty (CV) is two to ten times higher than national uncertainty; and (5) demonstrates that CEADs and MEIC yield consistent estimates across nine representative provinces. At the national scale, CAMS exhibits the smallest deviation from the National Greenhouse Gas Inventory (NGHGI), while ODIAC aligns most closely with the six-inventory mean during the study period. These clarifications have been added to Section 1 to highlight the study's novelty and rationale for using the latest inventory versions, and to Section 4 to summarize the new insights contributions.

**Revision:**

(1) **Section 1, paragraph 5:** "To this aim, this study conducts a comprehensive analysis of the spatiotemporal variation of China's anthropogenic CO2 emissions and investigates the differences among six widely used emission inventories at their latest versions: the global inventories ODIAC, EDGAR, MEIC, GEMS, CAMS, and the China-specific inventory CEADs. The data and methods

are presented in Section 2. We report our results in Section 3 and conclude the paper in Section 4. Compared with previous studies (Han et al., 2020b; Zheng et al., 2025), we extend the temporal coverage to 2000-2023, enabling a more current and consistent assessment of recent emission trends, inter-inventory discrepancies, and scale-dependent uncertainties across China."

(2) Section 4, paragraph 5: "In summary, this study extends previous work by identifying a three-phase trend in China's anthropogenic CO2 emissions from 2000 to 2023 and quantifying the emission uncertainties (1 \sigma) at both national and provincial levels. At the national level, CAMS shows the closest agreement with the government-reported NGHGI, while ODIAC aligns best with the multi-inventory mean over the study period. At the provincial level, the Chinese local inventories, CEADs and MEIC, provide the most consistent estimates for regional studies. Differences in spatial proxies significantly affect the spatial distribution of sectoral emissions, as shown by the contrasting transport emission patterns in EDGAR and MEIC. We also clarify the appropriate use of CEADs for provincial analyses. Our results further underscore the importance of improving the consistency of regional inventories to provide a stronger scientific basis for China's emission mitigation and carbon neutrality policies."

**2. Recommendations for users**

The conclusion clearly summarises findings but could be strengthened by adding actionable guidance. Readers would benefit from answers to the following questions:

- Which inventories are most reliable for specific applications?
- What are the main uncertainties that remain?
- How can inventory producers improve the next inventory versions?

A summary table of findings and recommendations could make this section more impactful.

**Response:** We thank the reviewer for these constructive suggestions. Providing practical recommendations would strengthen the manuscript, and we have revised the text to improve clarity for readers.

**(1) Consistency assessment at national and provincial levels**

We agree that identifying which inventories are more reliable is crucial. However, determining the absolute accuracy of each inventory requires direct comparison with independent observations (e.g., atmospheric CO2 measurements together with an inversion model), which is beyond the scope of this study. Therefore, in this study, we mainly assessed the internal consistency of the six inventories and their deviations from independent references. Specifically, we included the National Greenhouse Gas Inventory (NGHGI) reported by the Chinese government to the UNFCCC for national-level comparison. Our results show that CAMS exhibits the greatest consistency with the NGHGI, while ODIAC aligns most closely with the six-inventory mean. At the provincial level, uncertainties are 2-10 times higher than at the national scale. Although absolute references remain uncertain, CEADs and MEIC demonstrate strong agreement across nine representative provinces, particularly in Inner Mongolia, Shandong, Henan, Hubei, and Shanghai.

**(2) Main source of uncertainties**

Different downscale methods and spatial proxies might be the primary source of uncertainties across inventories. This is quantitatively supported by our finding that the uncertainties at provincial level are two to ten times higher than at the national level. Furthermore, our analysis shows that

differences in spatial proxies significantly affect the spatial distribution of sectoral emissions, as shown by the contrasting transport emission patterns in EDGAR and MEIC.

**(3) Recommendations to improve inventory reliability**

To enhance the reliability of future inventory versions, we recommend enhanced cross-validation with national statistics and transparent documentation of proxy methodologies. In addition, expanding ground-based and satellite observations would enable comprehensive independent validation. CO2 flux measurements can be directly compared with bottom-up estimates, while atmospheric CO2 mole fractions measurements, when integrated with inversion model, yield top-down emission estimates. These top-down results can then be systematically compared with bottom-up inventories to identify discrepancies across regional and national scales.

**Revision:**

[revised manuscript text omitted]

**3. Comparison to observations**

The study compares inventories against each other. Without observational benchmarks, it is difficult to assess which inventory is closer to reality. Could you explain why observational comparisons were not included? If data limitations prevented this, could you state them explicitly and discuss implications for interpreting results?

**Response:** We appreciate the reviewer's insightful comment. We fully agree that observational benchmarks are essential for evaluating the accuracy of emission inventories. However, such comparisons were not included in this study due to data limitations. Direct CO2 flux measurements, such as those from eddy covariance or mass balance, are spatially sparse and only represent local scales. Consequently, they are unsuitable for evaluating national or provincial emission totals. In

addition, fluxes derived from atmospheric inversion model together with CO2 mole fraction measurements can provide valuable top-down constraints but are also strongly affected by available data and model uncertainty. Therefore, incorporating these datasets would not provide consistent national-scale evaluation between 2000-2023.

Our study focuses on assessing the internal consistency among inventories and their deviations from independent references (i.e., the National Greenhouse Gas Inventory, NGHGI). We acknowledge that the absence of an observational benchmark limits the ability to identify which inventory is closer to reality. Future work should integrate direct flux observations and top-down emissions from inversion modeling to independently evaluate and constrain bottom-up inventories at both regional and national scales. We have added a clarification in Section 4, paragraph 6 to address this point.

**Revision:**

Section 4, paragraph 6: "Overall, reliable emissions quantification requires scale-appropriate inventories (e.g., the sectoral CEADs emissions versus the province-based CEADs emissions), improved spatial proxies (e.g., CPED vs. CARMA), and ensemble approaches to mitigate biases, especially in the carbon-intensive eastern regions. It should be noted that this study lacks an observational benchmark to assess these inventories. Future efforts should incorporate direct flux measurements or top-down emissions derived from inversion modeling, in combination with CO2 mole fraction observations, to compare and constrain bottom-up inventories at both regional and national scales."

**Specific comments**

**1. Line 80: MEIC is described as China-specific but later implied to be global. Could you clarify?**

Response: We thank the reviewer for pointing out this potential confusion regarding the MEIC inventory. We acknowledge that the distinction between MEIC's China-specific and global products was not sufficiently clarified. The MEIC team produces two distinct CO2 emission products: a China-specific version (MEIC-China-CO2) and a global version (MEIC-Global-CO2). We selected the MEIC-Global-CO2 product v1.0 based on its two primary advantages: it offers a higher spatial resolution (0.1°×0.1°) compared to the then-latest MEIC-China-CO2 v1.4 (0.25°×0.25°), and its temporal coverage extends closer to the most recent years (1970–2023 vs 1970–2020). Importantly, while this product is globally scoped, the emissions calculation within the Chinese region retains the accuracy of a local inventory by using Chinese local energy statistics (from the China Energy Statistics Yearbook, CESY)) and emission factors (from the China Emission Accounts and Datasets, CEADs). We have revised content in Section 2.1, paragraph 3 to harmonize these descriptions and clarify that the global version was selected based on its superior technical specifications (spatial resolution and temporal coverage).

**Revision:**

**Section 2.1, paragraph 3:** "...MEIC uses the transportation network data from the China Digital Road Network Map (CDRM) to constrain the distribution of vehicle activity as well as population density, GDP, and land use for other sectors (Li et al., 2017a; Xu et al., 2024b). In this study, we

use the latest MEIC-Global-CO2 product (v1.0), which provides higher spatial resolution (0.1° × 0.1°) and longer temporal coverage (1970-2023) than the MEIC-China-CO2 product (v1.4; 0.25° × 0.25°, up to 2020). It's noteworthy that although MEIC-Global-CO2 is a global product, its emissions calculations for China continue to rely on local energy statistics (CESY) and emission factors (CEADs), ensuring consistency with domestic data while improving spatiotemporal details."

**2. Line 88: You mention standardising inventories to a common grid. Could this process introduce uncertainties? If so, could you quantify or acknowledge them?**

**Response:** We appreciate the reviewer's comment. The standardization to a common grid introduces negligible additional uncertainty in this study. National and provincial totals are not affected, as they were derived either directly from the original inventory products or by spatially masking and summing emissions on their native grids.

For the gridded comparison, we adopted the MEIC grid as the spatial reference. For ODIAC, originally provided at a 1 km  $\times$  1 km resolution, emissions were spatially aggregated by summing all sub-grid values within each  $0.1^{\circ} \times 0.1^{\circ}$  cell to match the reference resolution. This aggregation preserves the total emissions without introducing interpolation-related errors. For EDGAR, CAMS, GEMS, and MEIC, all of which have the same resolution  $(0.1^{\circ} \times 0.1^{\circ})$  and identical latitude—longitude extents, we applied a nearest-neighbour method to ensure exact grid alignment. This approach maintains the original emission magnitudes and prevents artificial spatial gradients. Therefore, the regridding and aggregation procedures do not substantially affect either the spatial distribution or the total emissions, and the associated uncertainties are considered negligible.

**3. Line 174: Each growth phase is described with justification based on context, except from the third phase. Could you explain why emissions increase again after 2016?**

**Response:** We thank the reviewer for this valuable suggestion. We have added an explanation for the renewed increase in CO2 emissions after 2016. According to Zhang et al. (2020), the rebound was mainly driven by renewed infrastructure investment and the recovery of industrial activity after 2016. These developments substantially increased electricity demand, which was largely met by coal-fired power generation. As a result, fossil fuel consumption rose again, and the mitigation effect of the cleaner energy mix weakened compared with the 2012–2015 period. These points have been incorporated into the revised Results and Conclusion sections.

**Revision:**

- (1) **Section 3.1, paragraph 3:** "...From 2016 to 2023, all inventories show increased  $CO_2$  emissions again, with a slower rate  $(0.30\pm0.016 \text{ Gt year-1})$  compared to the first phase. This rebound could be attributed to the expansion of infrastructure investment and the recovery of coal-based power generation, as the mitigation effect of the cleaner energy mix weakened after 2016 (Zhang et al., 2020)."
- (2) **Section 4, paragraph 2:** "..., and a renewed growth of  $0.30\pm0.016$  Gt year-1 (2016–2023), mainly related to infrastructure-driven energy demand and coal use recovery following 2016. ..."


Response: We thank the reviewer for this insightful comment. We examined CO2 emissions from CEADs (sectors) and CEADs (provinces) for Shanxi and found that the large discrepancy mainly originates from raw coal-related emissions, which is the dominant contributor to total emissions (Wei, 2022). As shown in the figure below, CO2 emissions from raw coal in CEADs (provinces) are on average 665 Mt year-1 higher than those in CEADs (sectors), resulting in an overall mean difference of 512 Mt year-1 between the two datasets. This indicates that inconsistencies in fuel-specific accounting, particularly for raw coal, are a key contributor to the provincial-level uncertainty. The detailed comparison has been added to the Supplementary Material, and we have clarified this in the revised manuscript.

**Revision:**

Section 3.3.1, paragraph 1: "...In contrast, the CEADs (sectors) closely matches the other five independent inventories (ODIAC, EDGAR, MEIC, CAMS and GEMS), with its mean emissions deviating by no more than 3.84 Mt year-1 from the average of the five inventories. The large discrepancy between CEADs (provinces) and CEADs (sectors) mainly originates from the much higher raw coal-related emissions in CEADs (provinces) (Fig. S3), as coal is the dominant contributor to total emissions (Wei, 2022)."

**Section 7, Figure S3:**

Figure S3. Comparison of total CO2 emissions and raw coal—related CO2 emissions in Shanxi from CEADs (sectors) and CEADs (provinces) during 2000–2020. Solid lines represent total emissions, while dashed lines indicate emissions from raw coal combustion.

**Section 7, Figure S6:**

Figure S6. Comparison of national CO2 emissions from different versions of the EDGAR and MEIC inventories. The older versions (EDGAR v4.3.2 and MEIC-China-CO2 v1.3) used in Han et al. (2020b) are compared with the updated versions (EDGAR 2024 and MEIC-Global-CO2 v1.0) used in this study.

**12. Line 388: "Ensemble approaches" please define how this method would be used and explain why they would help mitigate biases**

Response: We thank the reviewer for this insightful comment. The term ensemble approaches in this study refers to statistical and model-based frameworks that integrate multiple emission inventories and auxiliary datasets (e.g., energy statistics, spatial proxies, and inversion-based flux estimates) to produce a consensus estimate and quantify uncertainty. Such approaches can take various forms, including weighted averaging, Bayesian inversion, or ensemble learning in machine learning. By combining independent datasets with different methodological assumptions and spatial representations, ensemble techniques reduce the influence of biases or errors present in any single inventory and provide more robust emission estimates.